# FEDGO: FEDERATED ENSEMBLE DISTILLATION WITH GAN-BASED OPTIMALITY

## ABSTRACT

For federated learning in practical settings, a significant challenge is the considerable diversity of data across clients. To tackle this data heterogeneity issue, it has been recognized that federated ensemble distillation is effective. Federated ensemble distillation requires an unlabeled dataset on the server, which could either be an extra dataset the server already possesses or a dataset generated by training a generator through a data-free approach. Then, it proceeds by generating pseudo-labels for the unlabeled data based on the predictions of client models and training the server model using this pseudo-labeled dataset. Consequently, the efficacy of ensemble distillation hinges on the quality of these pseudo-labels, which, in turn, poses a challenge of appropriately assigning weights to client predictions for each data point, particularly in scenarios with data heterogeneity. In this work, we suggest a provably near-optimal weighting method for federated ensemble distillation, inspired by theoretical results in generative adversarial networks (GANs). Our weighting method utilizes client discriminators, trained at the clients based on a generator distributed from the server and their own datasets. Our comprehensive experiments on various image classification tasks illustrate that our method significantly improves the performance over baselines, under various scenarios with and without extra server dataset. Furthermore, we provide an extensive analysis of additional communication cost, privacy leakage, and computational burden caused by our weighting method.

## 1 INTRODUCTION

Federated learning (FL) (McMahan et al., 2017) has received substantial attention in both industry and academia as a promising distributed learning approach. It enables numerous clients to collaboratively train a global model without sharing their private data. A major concern in deploying FL in practice is the severe data heterogeneity across clients. In the real world, it's probable that clients possess non-IID (identical and independently distributed) data distributions. It is known that the data heterogeneity results in unstable convergence and performance degradation (Li et al., 2019; Wang et al., 2020b; Li & Zhan, 2021; Kairouz et al., 2021; Huang et al., 2023; Karimireddy et al., 2020).

To address the data heterogeneity issue, various approaches have been taken, including regularizing the objectives of the client models (Karimireddy et al., 2020; Li et al., 2020; Liang et al., 2019; Yao et al., 2021; Mendieta et al., 2022), normalizing features or weights (Dong et al., 2022; Kim et al., 2023), utilizing past round models (Yao et al., 2021; Wang et al., 2023b), sharing feature information (Dai et al., 2023; Yang et al., 2024; Tang et al., 2024), introducing personalized layers (Huang et al., 2023), and learning the average input-output relation of client models through ensemble distillation (Chang et al., 2019; Gong et al., 2021; Deng et al., 2023; Sattler et al., 2020; Lin et al., 2020; Cho et al., 2022; Xing et al., 2022; Park et al., 2024; Wang et al., 2023a; Tang et al., 2022; Zhang et al., 2022; 2023b). In particular, the last approach, federated ensemble distillation, has recently gained significant attention for its effectiveness in mitigating data heterogeneity and for its advantage of being effectively applicable to heterogeneous client models. It requires an unlabeled dataset at the server, for which pseudo labels are created based on client predictions. By training on this pseudo-labeled dataset at the server, the server distills the knowledge from the clients. This additional dataset can be either public (Chang et al., 2019; Gong et al., 2021; Deng et al., 2023; Sattler et al., 2020), held only by the server due to its exceptional data collection capability (Lin et al., 2020; Cho et al., 2022; Xing et al., 2022; Park et al., 2024), or obtained through a data-free approach (Wang

Figure 1: A toy example of decision boundaries of aggregated models. Each point represents data, and its color represents the label. The background color represents the decision boundary of each model in the RGB channels. The oracle decision boundary, shown by the black lines, corresponds to the $x$-axis and $y$-axis. For aggregated models, we consider the parameter-averaged model (McMahan et al., 2017) and ensemble-distilled models using uniform weighting (Lin et al., 2020), variance weighting (Cho et al., 2022), entropy weighting (Deng et al., 2023; Park et al., 2024), domain-aware weighting (Wang et al., 2023a), and ours. Detailed settings are provided in Appendix E.1.

et al., 2023a; Tang et al., 2022; Zhang et al., 2022; 2023b). Note that the performance of ensemble distillation depends on the quality of the pseudo-labels, which ultimately translates into a problem of appropriately assigning weights to client predictions for each data point, particularly in situations of data heterogeneity. In this research stream of federated ensemble distillation, early studies like FedDF (Lin et al., 2020) applied uniform weighting. Subsequently, algorithms such as Fed-ET (Cho et al., 2022), FedHKT (Deng et al., 2023), FedDS (Park et al., 2024), and DaFKD (Wang et al., 2023a) emerged, which utilize metrics like variance, entropy, and judgement of client discriminator as indicators of confidence in client predictions for weighting. However, analysis regarding the rationale behind optimal weighting remains scarce.

In this paper, we suggest a novel weighting method for federated ensemble distillation that outperforms previous methods (Fig. 1), with theoretically justified optimality based on some results in generative adversarial networks (GANs) (Goodfellow et al., 2014). Our main contributions are summarized in the following:

- We propose **FedGO**: **F**ederated **E**nsemble **D**istillation with **G**AN-based **O**ptimality. Our algorithm incorporates a novel weighting method using the client discriminators that are trained at the clients based on the generator distributed from the server.

- The optimality of our proposed weighting method is theoretically justified. We define an optimal model ensemble and show that a knowledge-distilled model from an optimal model ensemble achieves the optimal performance, within an inherent gap due to the difference between the spanned hypothesis class of ensemble model and the hypothesis class of a single model. Then, based on the theoretical result for vanilla GAN (Goodfellow et al., 2014), we show that our weighting method using client discriminators constitutes an optimal model ensemble.

- We experimentally demonstrate significant improvements of FedGO over existing research both in final performance and convergence speed on multiple image datasets (CIFAR-10/100, ImageNet100). In particular, we demonstrate performance across various scenarios, including cases where the server holds an unlabeled dataset different from the client datasets and where the the server does not hold an unlabeled dataset and hence some data-free approaches are taken. Furthermore, we provide a comprehensive analysis of communication cost, privacy leakage, and computational burden for the proposed method.

## 2 SYSTEM MODEL AND RELATED WORK

**Federated Learning** In federated learning, the goal is to cooperatively train a global model based on data distributed among $K$ clients, by exchanging the models between a server and the clients.

We focus on classification tasks in this paper. Let $\mathcal{X}$ denote the data domain and $y$ denote the labeling function that outputs the label of the data $x \in \mathcal{X}$. A model $f(\cdot; \theta)$ is parameterized by $\theta \in \Theta$ where $\Theta$ is the set of model parameters and $\mathcal{H} = \{h | h(\cdot) = f(\cdot; \theta), \theta \in \Theta\}$ denotes the class of parameterized models. For a distribution $q$ on $\mathcal{X}$, $h_q^*$ denotes the expected loss minimizer on $q$, i.e., $h_q^* \triangleq \arg\min_{h \in \mathcal{H}} \mathcal{L}_q(h)$, where $\mathcal{L}_q(h) = \mathbf{E}_q[l(h(x), y(x))]$ and $l$ is the loss function. Client

$k$ possesses a (labeled) dataset $S_k$ of $n_k$ data points, sampled over $\mathcal{X}$ i.i.d. according to distribution $p_k$. Then $p = \sum_{k=1}^K \pi_k \cdot p_k$, where $\pi_k = \frac{n_k}{\sum_{k'=1}^K n_{k'}}$, is the average of client data distribution. The objective of federated learning is given as follows:

$$\min_{h \in \mathcal{H}} \mathcal{L}_p(h) = \min_{h \in \mathcal{H}} \mathbf{E}_p[l(h(x), y(x))] \tag{1}$$

$$= \min_{h \in \mathcal{H}} \sum_{k=1}^K \pi_k \cdot \mathbf{E}_{p_k}[l(h(x), y(x))] = \min_{h \in \mathcal{H}} \sum_{k=1}^K \pi_k \cdot \mathcal{L}_{p_k}(h). \tag{2}$$

In each communication round $t$, a subset $A^t$ of clients downloads the current server model and trains it based on $S_k$ with the objective of minimizing $\mathcal{L}_{p_k}(h)$. Then it sends the trained model to the server. The server aggregates these client models to update the server model. The aforementioned procedure is repeated at the next communication round. For the aggregation of client models at the server, the FedAVG algorithm (McMahan et al., 2017) constructs the server model with parameter $\theta_s^t$ for round $t$ as the average of model parameters $\theta_k^t$ for $k \in A^t$ received in round $t$ (line 7 of Algorithm 1). When the client data distributions are homogeneous, each $p_k$ is same as $p$ and hence $\mathcal{L}_{p_k}$ becomes same as $\mathcal{L}_p$. However, when the client data distributions are heterogeneous, $\mathcal{L}_{p_k}$ and $\mathcal{L}_p$ are not same, leading to a significant degradation in the convergence rate of FedAVG to the global optimum (Li et al., 2019).

In the following, we introduce federated ensemble distillation using an unlabeled dataset on the server to address client data heterogeneity.

**Federated Ensemble Distillation**  To address client data heterogeneity, there has been a line of research on federated ensemble distillation using an unlabeled dataset on the server. This unlabeled dataset may either be available from the outset (Lin et al., 2020; Cho et al., 2022; Deng et al., 2023; Park et al., 2024) or produced through a generator trained as a part of FL by taking a data-free approach (Rasouli et al., 2020; Guerraoui et al., 2020; Li et al., 2022; Wang et al., 2023c; Fan & Liu, 2020; Behera et al., 2022; Hardy et al., 2019; Xiong et al., 2023; Zhang et al., 2021; 2023a; Wang et al., 2023a; Zhang et al., 2022; 2023b). With the unlabeled dataset, the server model undergoes additional training to learn the average input-output relationship of client models.

Algorithm 1 describes this federated ensemble distillation, when the client and server model structures are the same. Here $\sigma$ represents the softmax function, and KL denotes the Kullback-Leibler divergence. If the model output already includes the softmax activation, then the softmax function is omitted in lines 10 and 11. After averaging client model parameters in line 7, the performance of the server model depends on the quality of the pseudo-labels, as the server model undergoes additional training with those pseudo-labels. Moreover, the quality of pseudo-labels $\tilde{y}(\cdot)$ relies on designing the weighting function $w_k(\cdot)$, which determines the weighting of client $k$'s output. Therefore, designing a better-performing ensemble distillation during the server update ultimately boils down to designing a better-performing weighting function.

For the weighting function, FedDF (Lin et al., 2020) uses uniform weights for each client, i.e., $w_k(x) = \frac{1}{|A^t|}$ for all $k$ in $A^t$. Subsequently, algorithms assigning higher weights to the outputs of more confident clients have been proposed. In Fed-ET (Cho et al., 2022), higher weights are assigned to models with larger output logit variance, i.e., $w_k(x) = \frac{\text{Var}(f(x; \theta_k^t))}{\sum_{i \in A^t} \text{Var}(f(x; \theta_i^t))}$. FedHKT (Deng et al., 2023) and FedDS (Park et al., 2024) allocate higher weights to models with smaller output softmax entropy, i.e., $w_k(x) = \frac{\exp(-\text{Entropy}(\sigma(f(x; \theta_k^t)))/\tau)}{\sum_{i \in A^t} \exp(-\text{Entropy}(\sigma(f(x; \theta_i^t)))/\tau)}$, where $\tau$ is the temperature parameter. In DaFKD (Wang et al., 2023a), while training a global generator and client discriminators at each round, ensemble distillation is performed on unlabeled dataset generated by the global generator by assigning higher weights to models with larger discriminator outputs, i.e., $w_k(x) = \frac{D_k^t(x)}{\sum_{i \in A^t} D_i^t(x)}$ where $D_k^t$ is the client $k$'s discriminator against the global generator at round $t$.

For theoretical aspects, generalization bounds of an ensemble model are presented in Lin et al. (2020); Cho et al. (2022); Wang et al. (2023a) for a binary classification task under $\ell_1$ loss. For $n_k = \frac{n}{K}$ for all $k$, a generalization bound for an ensemble model with fixed weights $\alpha_1, ..., \alpha_K$ with $\sum_k \alpha_k = 1$ is given as follows (Lin et al., 2020; Cho et al., 2022): for any $\delta \in (0, 1)$, the following

---

**Algorithm 1** Federated learning with $K$ clients for $T$ communication rounds, with ensemble distillation exploiting unlabeled dataset on the server. Client $k$ possesses $n_k$ data points, and the fraction $C$ of clients participate in each communication round. $f(\cdot; \theta)$ stands for the model with parameter $\theta$, and $\mu$ stands for the step size.

---

**Require:** Client labeled dataset $\{S_k\}_{k=1}^K$, server unlabeled dataset $U$
1: Initialize server model $f(\cdot, \theta_s^0)$ with parameter $\theta_s^0$
2: **for** communication round $t = 1$ to $T$ **do**
3:      $A^t \leftarrow$ sample $\lfloor C \cdot K \rfloor$ clients
4:      **parfor** client $k \in A^t$ **do**
5:          $\theta_k^t \leftarrow ClientUpdate(\theta_s^{t-1}, S_k)$               $\triangleright$ Gradient update $\theta_s^{t-1}$ with $S_k$
6:      **end parfor**
7:      $\theta_s^t \leftarrow \sum_{k \in A^t} \frac{n_k}{\sum_{i \in A^t} n_i} \cdot \theta_k^t$
8:      **for** server train epoch $e = 1$ to $E_s$ **do**
9:          **for** unlabeled minibatch $u \in U$ **do**
10:             $\tilde{y}(u) \leftarrow \sigma(\sum_{k \in A^t} w_k(u) \cdot f(u; \theta_k^t))$ $\triangleright$ Label as a weighted sum of client predictions
11:             $\theta_s^t \leftarrow \theta_s^t - \mu \cdot \nabla_{\theta_s^t} \mathrm{KL}(\tilde{y}(u), \sigma(f(u; \theta_s^t)))$            $\triangleright$ Ensemble distillation
12:          **end for**
13:      **end for**
14: **end for**
15: **return** $f(\cdot, \theta_s^T)$

---

holds with probability $1 - \delta$:

$$\mathcal{L}_p(\sum_{k=1}^K \alpha_k h_{\hat{p}_k}^*) \le \sum_{k=1}^K \alpha_k \cdot \left[ \mathcal{L}_{\hat{p}_k}(h_{\hat{p}_k}^*) + \frac{1}{2} d_{\mathcal{H} \triangle \mathcal{H}}(p_k, p) + \lambda_k + O\left( \frac{\log(\delta^{-1})}{\sqrt{n_k}} \right) \right]. \qquad (3)$$

Here, $\hat{p}_k$ is the empirical distribution by sampling $n_k$ data points i.i.d. according to $p_k$, $d_{\mathcal{H} \triangle \mathcal{H}}$ denotes the discrepancy between two distributions, $\lambda_k = \inf_h \mathcal{L}_{p_k}(h) + \mathcal{L}_p(h)$, and $\tau_{\mathcal{H}}$ is growth function bounded by polynomial of the VC-dimension of $\mathcal{H}$.

On the other hand, a generalization bound for a weighted ensemble model with weight function $w_k(x) = \frac{D_k^t(x)}{\sum_{i \in A^t} D_i^t(x))}$ is given as follows (Wang et al., 2023a): for any $\delta \in (0, 1)$ and $\sigma > 0$, the following holds with probability $1 - \delta$:

$$\mathcal{L}_p(\sum_{k=1}^K w_k \cdot h_{\hat{p}_k}^*) \le (K + 1) \cdot \sum_{k=1}^K \frac{1}{K} \cdot \left[ \mathcal{L}_{\hat{p}_k}(h_{\hat{p}_k}^*) + \sqrt{\frac{\sigma^2 \log \frac{2K}{\delta}}{2 n_k}} \right]. \qquad (4)$$

The above bounds relate the loss of an ensemble model (the LHS of (3) and (4)) to the average empirical loss of client models (the first term in the RHS of (3) and (4)). The proofs of these bounds rely on the results in domain adaptation theory for binary classification [Shalev-Shwartz & Ben-David, Theorem 6.11; Ben-David et al., Lemma 3]. Note that the bound (3) assumes a fixed weight per client irrelevant to data points, hence there is a lack of analysis for assigning varying weights per data point. The bound (4) assumes a specific weighting function of each data point, but it is too loose because it becomes vacuous as $K$ increases. Consequently, guidance on determining appropriate weights for each client per data point is limited. Moreover, in federated ensemble distillation, our ultimate interest is in the loss of the server model, knowledge-distilled from the ensemble model. Note that the hypothesis class of ensemble models is in general larger than that of single models, and hence there exists an inherent gap between the losses of an ensemble model and the knowledge-distilled model. However, the above bounds do not provide an analysis on this gap.

In Section 3.1, we define an optimal model ensemble and show that the server model knowledge-distilled from an optimal model ensemble achieves the optimal loss within the gap arising from the distillation step, which depends on the inherent difference between the hypothesis classes of the server model and the ensemble model, along with the distribution discrepancy between the average client distribution $p$ and the distribution $p_s$ of unlabeled data on the server.

**Generative Adversarial Network** The generative adversarial networks (GANs) are a class of powerful generative models composed of a generator and a discriminator (Goodfellow et al., 2014;

Gulrajani et al., 2017; Radford et al., 2015; Chen et al., 2016; Zhu et al., 2017; Choi et al., 2018; Karras et al., 2019). They are trained in an unsupervised learning manner, requiring no class labels. The discriminator aims to distinguish between real images from the dataset and fake images generated by the generator. Meanwhile, the generator strives to produce images that can fool the discriminator.

In Goodfellow et al. (2014), the authors showed that the output of an optimal discriminator against a fixed generator can be expressed in terms of distributions of real and fake images.

**Theorem 1.** (Goodfellow et al., 2014, Proposition 1) For a fixed generator $G$, let $p_g$ and $p_{\text{data}}$ denote the density functions of the generated distribution by $G$ and the real data distribution, respectively. Then the output of an optimal discriminator $D$ for input data $x$ is given as follows:

$$D(x) = \frac{p_{\text{data}}(x)}{p_{\text{data}}(x) + p_g(x)}. \tag{5}$$

Using the above result, we develop a method of assigning weights to client predictions in Section 3.

## 3 PROPOSED METHOD

In this section, we propose a weighting method for federated ensemble distillation. First, theoretical results are presented in Section 3.1. In Section 3.1.1, we define an optimal model ensemble and give a bound on the loss of the server model knowledge-distilled from an optimal model ensemble. Next, in Section 3.1.2, we propose a client weighting method to construct an optimal model ensemble, based on Theorem 1. In Section 3.2, we introduce our FedGO algorithm, leveraging the theoretical results. We note that a generalization bound of an ensemble model with our proposed weighting method comparable with (3) is provided in Appendix C.

### 3.1 THEORETICAL RESULTS

#### 3.1.1 ENSEMBLE DISTILLATION WITH OPTIMAL MODEL ENSEMBLE

We first define an optimal model ensemble.

**Definition 1.** For $K$ clients, the ensemble of their models and weight functions $\{(h_k, w_k)\}_{k=1}^K$ is said to be an optimal model ensemble if the following holds:

$$\mathcal{L}_p\left(\sum_{k=1}^K w_k \cdot h_k\right) = \mathbf{E}_p\left[l\left(\sum_{k=1}^K w_k(x) \cdot h_k(x), y(x)\right)\right] \leq \min_{h \in \mathcal{H}} \mathcal{L}_p(h) = \mathcal{L}_p(h_p^*). \tag{6}$$

We remind that the objective of federated learning is to train a model that minimizes the expected loss over the average client distribution $p$ as shown in (1). If $\{(h_k, w_k)\}_{k=1}^K$ is an optimal model ensemble, its expected loss over $p$ is less than or equal to the minimum expected loss over $p$ achievable by a single model, i.e., $\min_{h \in \mathcal{H}} \mathcal{L}_p(h)$.

However, we cannot guarantee that a knowledge-distilled model from an optimal model ensemble would be optimal, i.e., achieve $\min_{h \in \mathcal{H}} \mathcal{L}_p(h)$, due to the following two reasons: 1) the ensemble model $\sum_{k=1}^K w_k \cdot h_k$ may lie outside the hypothesis class $\mathcal{H}$ of a single model and 2) the distribution used for knowledge distillation (the distribution $p_s$ of unlabeled data on the server) can be different from $p$. In the following theorem, we present a bound on the expected loss over $p$ of a single model by taking into account these factors. For two hypotheses $h, h' \in \mathcal{H}$ and a distribution $q$ over $\mathcal{X}$, the expected difference between $h$ and $h'$ over $q$, denoted $\mathcal{L}_q(h, h')$, is defined as follows:

$$\mathcal{L}_q(h, h') \triangleq \mathbf{E}_q\left[(l(h(x), h'(x))\right]. \tag{7}$$

**Theorem 2.** (Informal) Let $\bar{\mathcal{H}} \triangleq \{\sum_{k=1}^K w_k \cdot h_k | h_j \in \mathcal{H}, w_j : \mathcal{X} \to [0,1], \sum_{k=1}^K w_k(x) = 1, j = 1, \cdots, K, x \in \mathcal{X}\}$ be the spanned hypothesis class, $p_s$ be a distribution on $\mathcal{X}$, and $\{(h_k, w_k)\}_{k=1}^K$ be an ensemble of client models and weight functions. Then for any $h \in \mathcal{H}$, the following holds:

$$\mathcal{L}_p(h) \leq \mathcal{L}_p(\sum_{k=1}^K w_k \cdot h_k) + \mathcal{L}_{p_s}(h, \sum_{k=1}^K w_k \cdot h_k) + \frac{1}{2}d_{\bar{\mathcal{H}} \triangle \bar{\mathcal{H}}}(p, p_s). \tag{8}$$

The formal statement and the proof of the above theorem are in Appendix A. Let us provide a brief sketch of the proof. Utilizing the results from Ben-David et al. (2006) and Crammer et al. (2008), we have $\mathcal{L}_p(h) \leq \mathcal{L}_p(\sum_{k=1}^{K} w_k \cdot h_k) + \mathcal{L}_p(h, \sum_{k=1}^{K} w_k \cdot h_k)$. Then from the triangular inequality, we obtain $\mathcal{L}_p(h, \sum_{k=1}^{K} w_k \cdot h_k) \leq \mathcal{L}_{p_s}(h, \sum_{k=1}^{K} w_k \cdot h_k) + |\mathcal{L}_p(h, \sum_{k=1}^{K} w_k \cdot h_k) - \mathcal{L}_{p_s}(h, \sum_{k=1}^{K} w_k \cdot h_k)|$. Now the desired inequality is obtained by applying the results from (Ben-David et al., Lemma 3).

From Theorem 2, we can ascertain the following. The loss of the server model $h$ is bounded by the sum of three losses: 1) expected loss of the ensemble model over $p$, 2) difference between $h$ and $\sum_{k=1}^{K} w_k \cdot h_k$ over $p_s$, and 3) the distribution discrepancy between $p$ and $p_s$.

The following corollary is a direct consequence of Theorem 2 and Definition 1.

**Corollary 1.** (Informal) For an optimal model ensemble $\{(h_k, w_k)\}_{k=1}^{K}$, the following holds for any $h \in \mathcal{H}$:

$$\mathcal{L}_p(h_p^*) \leq \mathcal{L}_p(h) \leq \mathcal{L}_p(h_p^*) + \mathcal{L}_{p_s}(h, \sum_{k=1}^{K} w_k \cdot h_k) + \frac{1}{2} d_{\bar{\mathcal{H}} \triangle \bar{\mathcal{H}}}(p, p_s). \tag{9}$$

Corollary 1 demonstrates the powerfulness of an optimal model ensemble. If an optimal model ensemble is constituted, the difference between the expected loss of the server model over $p$ and the minimum expected loss $\mathcal{L}_p(h_p^*) = \min_{h \in \mathcal{H}} \mathcal{L}_p(h)$ is bounded by the distillation loss, which depends on the inherent difference between the hypothesis class $\mathcal{H}$ and the spanned hypothesis class $\bar{\mathcal{H}}$, along with the distribution discrepancy between $p$ and $p_s$.

In the next subsection, we propose a weighting method to constitute an optimal model ensemble.

### 3.1.2 CLIENT WEIGHTING FOR OPTIMAL MODEL ENSEMBLE

Let us assume that the server has models $\{h_{p_k}^*\}_{k=1}^{K}$ trained by clients based on their respective data distributions $\{p_k\}_{k=1}^{K}$. In the following theorem, we present weight functions $\{w_k\}_{k=1}^{K}$ such that the ensemble of $\{h_{p_k}^*, w_k\}_{k=1}^{K}$ constitutes an optimal model ensemble.

**Theorem 3.** Let the loss function $l$ be convex. Define the client weight functions $\{w_k^*\}_{k=1}^{K}$ as follows:

$$w_k^*(x) \triangleq \frac{n_k \cdot p_k(x)}{\sum_{i=1}^{K} n_i \cdot p_i(x)} = \frac{\pi_k \cdot p_k(x)}{\sum_{i=1}^{K} \pi_i \cdot p_i(x)}. \tag{10}$$

Then, the ensemble $\{h_{p_k}^*, w_k^*\}_{k=1}^{K}$ is an optimal model ensemble, i.e., $\mathcal{L}_p\left(\sum_k w_k^* \cdot h_{p_k}^*\right) \leq \mathcal{L}_p(h_p^*)$.

Theorem 3 follows from some manipulations based on the convexity of the loss and the definitions of $w_k^*$'s and $h_{p_k}^*$'s, and its full proof is provided in Appendix B.

Theorem 3 demonstrates that for data point $x$, weighting according to each client's proportion of having $x$ constitutes an optimal model ensemble. However, even if weighting each client according to Theorem 3 constitues an optimal model ensemble, it is not feasible without knowing the data distribution $p_k$ of each client. Theorem 4 addresses this issue based on Theorem 1 and provides hints on how to implement an optimal model ensemble.

**Definition 2.** (Odds): For $\phi \in (0, 1)$, its odds value $\Phi$ is defined as $\Phi(\phi) = \frac{\phi}{1-\phi}$.

**Theorem 4.** For a fixed generator $G$ with generating distribution $p_g$, let $D_k$ be an optimal discriminator for generator $G$ and client $k$'s distribution $p_k$. Assume that $D_k$ outputs a value over $(0, 1)$ using a sigmoid activation function, and let $\Phi_k(x) \triangleq \Phi(D_k(x))$. Then, for $x \in supp(p_g)$, the following holds:

$$\frac{n_k \cdot \Phi_k(x)}{\sum_{i=1}^{K} n_i \cdot \Phi_i(x)} = \frac{\pi_k \cdot p_k(x)}{\sum_{i=1}^{K} \pi_i \cdot p_i(x)} = w_k^*(x). \tag{11}$$

Theorem 4 is a direct consequence of Theorem 1, because $\Phi_k(x) = \frac{p_k(x)}{p_g(x)}$ from Theorem 1. Theorem 4 indicates that if the server once receives the optimal discriminators $\{D_k\}_{k=1}^{K}$ trained by the

clients, it can use those discriminators to calculate the weights for optimal model ensemble. Note that the generator $G$ only needs to generate a wide distribution capable of producing sufficiently diverse samples. Therefore, one can use an off-the-shelf generator pretrained on a large dataset.

## 3.2 PROPOSED ALGORITHM: FEDGO

By leveraging the theoretical results in Section 3.1, we propose FedGO that constitutes an optimal model ensemble and performs knowledge distillation. The main technical novelty of FedGO lies in implementing the optimal weighting function $w_k^*$ using client discriminators, which is a versatile technique that can be integrated to both the following scenarios with/without extra server dataset.

(S1) The server holds an extra unlabeled dataset.

(S2) The server holds no unlabeled dataset, thus a data-free approach is needed.

For completeness, let us describe how the FedGO algorithm can be adapted depending on the cases (S1) and (S2). FedGO largely consists of two stages: pre-FL and main-FL. In the pre-FL stage, the server and the clients exchange the generator and the discriminators. First, the server obtains a generator through one of the following three methods, and distributes the generator to the clients.

(G1) Train a generator with an unlabeled dataset on the server, which is possible under (S1).

(G2) Load an off-the-shelf generator pretrained on a sufficiently rich dataset.

(G3) Train a generator through an FL approach, e.g., using FedGAN (Rasouli et al., 2020).

After receiving the generator, each client trains its own discriminator based on its dataset and sends the discriminator to the server.

The main-FL stage operates according to Algorithm 1, except that the server assigns weights for pseudo-labeling according to (11) using the client discriminators. For the server unlabeled dataset $U$ used for distillation, which we call distillation dataset, we consider the following cases:

(D1) Use the same dataset held by the server, which is possible under (S1).

(D2) Produce a distillation dataset using the generator from (G2).

(D3) Produce a distillation dataset using the generator from (G3).

A comprehensive analysis of additional communication cost, privacy leakage, and computational burden according to the methods for obtaining the generator and distillation set is provided in Table 1, which shows the trade-off among the methods. In particular, an extra dataset at the server makes the communication cost and the client-side privacy and computational burden negligible, at the expense of server-side privacy leakage. In the absence of server dataset, the use of an off-the-shelf generator makes all the burdens negligible, but it can be challenging to secure an off-the-shelf generator whose generation distribution is similar to the client data distribution. Lastly, the data-free approach (G3)+(D3) does not require an extra server dataset or an external generator, but it increases the communication burden and the privacy and computational burden on the client side.

A detailed description of FedGO and explanation for Table 1 can be found in Appendices D and G, respectively.

Table 1: A comprehensive analysis of additional communication burden, privacy leakage, and computational burden caused by the proposed weighting method, compared to FedAVG.

| Extra Server Dataset | Generator Preparation | Distillation Dataset | Communication Cost | Privacy Leakage | | Client-side Computational Burden |
|---|---|---|---|---|---|---|
| | | | | Server-side | Client-side | |
| S1 | G1 | D1 | Negligible | Non-negligible | Negligible | Negligible |
| S1 | G2 | D1 | Negligible | Non-negligible | Negligible | Negligible |
| S2 | G2 | D2 | Negligible | - | Negligible | Negligible |
| S2 | G3 | D3 | Non-negligible | - | Non-negligible | Non-negligible |

## 4 EXPERIMENTAL RESULTS

In this section, we present the experimental results. All experimental results were obtained using five different random seeds, and the reported results are presented as the mean $\pm$ standard deviation.

### 4.1 EXPERIMENTAL SETTING

**Datasets and FL Setup** We employed datasets CIFAR-10/100 (Krizhevsky et al., 2009) (MIT license) and downsampled ImageNet100 (ImageNet100 dataset; Chrabaszcz et al., 2017). Unless specified otherwise, the entire client dataset corresponds to half of the specified client dataset (half for each class), and each client dataset is sampled from the entire client dataset according to Dirichlet($\alpha$), akin to setups in Lin et al. (2020); Cho et al. (2022). $\alpha$ is set to 0.1 and 0.05 to represent data-heterogeneous scenarios. The server dataset corresponds to half of the specified server dataset (half for each class) without labels. If not otherwise specified, the server dataset and the client datasets partition the same dataset disjointly. We considered 20 and 100 clients (20 clients if not specified otherwise), assuming that 40% of the clients participate in each communication round.

**Models and Baselines** For architecture, we employed ResNet-18 (He et al., 2016) with batch normalization layers (Ioffe & Szegedy, 2015). For baselines, we considered the vanilla FedAVG (McMahan et al., 2017) and FedProx Li et al. (2020) that do not perform ensemble distillation, FedDF (Lin et al., 2020), FedGKD$^+$ (Yao et al., 2021) and DaFKD (Wang et al., 2023a) that incorporate ensemble distillation. For comparison with other weighting methods, we considered the variance-based weighting method of Cho et al. (2022), the entropy-based methods of Deng et al. (2023) and Park et al. (2024), and the domain-aware method of Wang et al. (2023a), described in Section 2. As an upper bound of the performance, we also compared with central training that trains the server model directly using the entire client dataset. FedGO and DaFKD require image generators and discriminators. For the generator, we considered the three approaches (G1), (G2), and (G3) in Section 3.2. For (G1) and (G3), we adopted the model architecture and training method proposed in WGAN-GP (Gulrajani et al., 2017). For (G2), we employed StyleGAN-XL (Sauer et al., 2022), pretrained on ImageNet (Krizhevsky et al., 2012). Unless specified otherwise, we assume (G1). For discriminators, we utilized a 4-layer CNN. More experimental details are provided in Appendix E.2.

### 4.2 RESULTS

**Test Accuracy and Convergence Speed** Table 2 shows the test accuracy of the server model and Table 3 presents the communication rounds required for the server model to achieve target accuracy ($\text{Acc}_{\text{target}}$) for the first time, for the baselines and FedGO, on CIFAR-10/100 and ImageNet100 datasets. Our FedGO algorithm exhibits the smallest performance gap from the central training and the fastest convergence speed across all the datasets and data heterogeneity settings.

For CIFAR-10 with $\alpha = 0.1$, our FedGO algorithm shows a performance improvement of over 7%p compared to the baselines. However, we observe a diminishing gain for CIFAR-100 and ImageNet100. We argue in Appendix F.1 that this is not due to the marginal improvement in FedGO's ensemble performance, but rather due to larger distillation loss as the server model more struggles to keep up with the performance of the ensemble model.

**Comparison of Weighting Methods** Figure 2 shows the ensemble test accuracy along with communication rounds on the CIFAR-10 dataset, according to weighting methods. We evaluated ensemble test accuracy to compare the efficacy of each method in generating pseudo-labels. For the baseline weighting methods, we considered the uniform (Lin et al., 2020), the variance-based (Cho et al., 2022), the entropy-based (Deng et al., 2023; Park et al., 2024), and the domain-aware (Wang et al., 2023a) methods. For fair comparison, all the baselines follow the same steps except the weighting methods. The effectiveness of our weighting method is demonstrated by its ensemble test accuracy outperforming all the other weighting methods over all communication rounds.

**Results with 100 Clients** Figure 3 shows (a) the test accuracy of the server model, (b) the test accuracy of the ensemble model, and (c) the test loss of the ensemble model during the training process for $K = 100$ clients on CIFAR-10 dataset with $\alpha = 0.05$. The latter two measures were evaluated only for algorithms incorporating ensemble distillation. FedGO achieves the test accuracy

Table 2: Server test accuracy (%) of our FedGO and baselines on three image datasets at the 100-th communication round. A smaller $\alpha$ indicates higher heterogeneity.

| | CIFAR-10 | | CIFAR-100 | | ImageNet100 | |
|---|---|---|---|---|---|---|
| | $\alpha = 0.1$ | $\alpha = 0.05$ | $\alpha = 0.1$ | $\alpha = 0.05$ | $\alpha = 0.1$ | $\alpha = 0.05$ |
| Central Training | 85.33±0.25 | | 51.72±0.65 | | 43.20±1.00 | |
| FedAVG | 58.65±5.75 | 46.61±8.54 | 38.93±0.74 | 36.66±0.97 | 29.44±0.41 | 27.58±0.88 |
| FedProx | 64.69±2.15 | 55.56±9.86 | 38.21±0.95 | 34.44±1.26 | 29.96±0.66 | 26.99±0.97 |
| FedDF | 71.56±5.09 | 59.53±9.88 | 42.74±1.22 | 37.18±1.03 | 33.48±1.00 | 30.94±1.60 |
| FedGKD$^+$ | 72.59±4.10 | 59.96±8.60 | 43.35±1.14 | 40.47±1.00 | 34.10±0.67 | 31.42±0.93 |
| DaFKD | 71.52±5.56 | 67.51±10.77 | 44.12±2.25 | 39.50±0.85 | 33.34±0.69 | 31.59±1.46 |
| **FedGO (ours)** | **79.62**±4.36 | **72.35**±9.01 | **44.66**±1.27 | **41.04**±0.99 | **34.20**±0.71 | **31.70**±1.55 |

Table 3: The number of communication rounds to achieve a test accuracy of at least Acc$_{target}$.

| | CIFAR-10 | | CIFAR-100 | | ImageNet100 | |
|---|---|---|---|---|---|---|
| | $\alpha = 0.1$ | $\alpha = 0.05$ | $\alpha = 0.1$ | $\alpha = 0.05$ | $\alpha = 0.1$ | $\alpha = 0.05$ |
| Acc$_{target}$ | 60% | 45% | 35% | 35% | 25% | 25% |
| FedAVG | 65.6±22.8 | 47.4±14.9 | 42.4±12.8 | 76.0±8.5 | 22.2±3.1 | 43.8±7.3 |
| FedProx | 38.0±9.1 | 33.0±12.7 | 45.6±5.9 | 86.0±11.8 | 20.8±3.8 | 47.6±5.8 |
| FedDF | 5.4±1.4 | 6.0±1.5 | 15.2±5.7 | 78.0±23.8 | 9.4±1.9 | 22.0±5.7 |
| FedGKD$^+$ | 5.6±1.6 | 4.2±1.2 | 12.6±3.3 | 39.8±19.6 | 9.0±1.4 | 14.8±2.5 |
| DaFKD | 5.6±1.4 | 3.0±0.6 | 13.4±5.4 | 50.2±27.9 | 9.0±2.8 | 15.6±4.1 |
| **FedGO (ours)** | **3.0**±0.9 | **2.0**±0.6 | **11.0**±2.1 | **25.4**±9.1 | **8.4**±1.0 | **12.6**±1.6 |

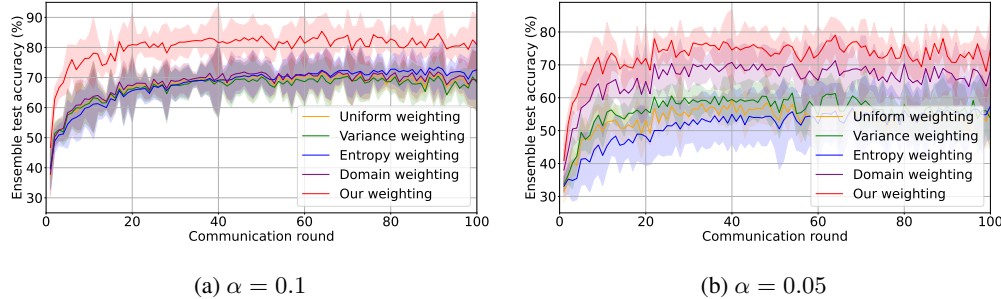

(a) $\alpha = 0.1$         (b) $\alpha = 0.05$

Figure 2: Ensemble test accuracy (%) of FedGO and other baseline weighting methods over communication rounds on CIFAR-10 with $\alpha = 0.1$ and $\alpha = 0.05$.

of 69.52%, which is slightly lower than 72.35% with 20 clients (Table 2). In comparison, FedAVG, FedProx, FedDF, FedGKD$^+$, and DaFKD show significant performance drops to 33.40%, 35.07%, 44.36%, 45.44%, and 59.62%, respectively. This demonstrates that even in settings with a large number of clients, FedGO exhibits robust performance compared to the baselines.

In terms of the test accuracy and the test loss of the ensemble model, FedGO consistently demonstrates superior performance across all rounds compared to the baseline algorithms. Furthermore, unlike the baseline algorithms, whose test loss initially decreases but then becomes unstable and increases from early rounds, FedGO's loss converges with small deviation.

**FedGO with a Pretrained Generator** If there exists a pretrained generator capable of generating sufficiently diverse data, the server can distribute the pretrained generator to clients instead of training a generator from scratch using the server's unlabeled dataset, which corresponds to the case (G2) in Section 3.2. This approach has the advantage of saving the server's computing resources required for training a generator.

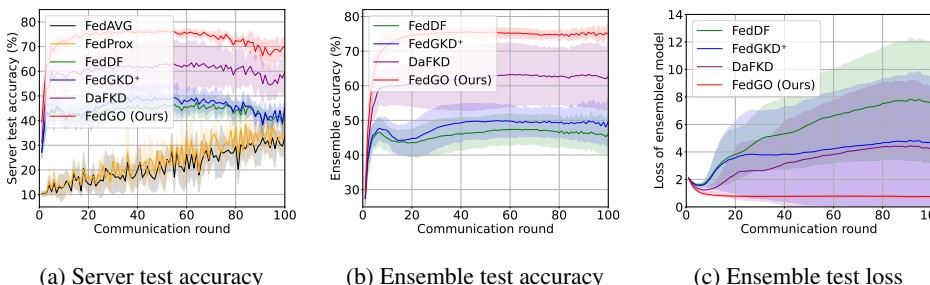

(a) Server test accuracy (b) Ensemble test accuracy (c) Ensemble test loss

Figure 3: Server test accuracy (%), test accuracy of the ensemble model (%), and test loss of the ensemble model of our FedGO and baselines for 100 clients on CIFAR-10 dataset with $\alpha = 0.05$.

Table 4 reports the performance of FedGO for various datasets with $\alpha = 0.05$, when using a generator trained with the server's unlabeled dataset versus using a generator pretrained on ImageNet (Krizhevsky et al., 2012). We observe that utilizing the pretrained generator results in superior performance on CIFAR-10 and ImageNet100, whereas it remains the same for CIFAR-100. A key factor contributing to performance enhancement seems to be the larger model structure of the pretrained generator and its training with a richer dataset. This enhances the generalization performance of client discriminators, enabling optimal weighting even for test data. However, since the assumption of Theorem 4 does not hold for $x \in \mathrm{supp}(p) \setminus \mathrm{supp}(p_g)$, the portion of data for which an optimal weighting is guaranteed decreases as the portion of $p$'s support not covered by $p_g$ increases, potentially leading to performance degradation. We note that ImageNet100 is a subset of ImageNet, and ImageNet includes the classes of CIFAR-10 except deer. However, there are several classes of CIFAR-100 not included in ImageNet, which could possibly result in no performance gain.

Table 4: Server test accuracy (%) of our FedGO with a generator trained with the unlabeled dataset on the server (Scratch) and with an off-the-shelf generator pretrained on ImageNet (Pretrained) on three image datasets with $\alpha = 0.05$.

| | CIFAR-10 | | CIFAR-100 | | ImageNet100 | |
|---|---|---|---|---|---|---|
| Generator | Scratch | Pretrained | Scratch | Pretrained | Scratch | Pretrained |
| Accuracy | 72.35±9.01 | **74.40**±6.97 | **41.04**±0.99 | **41.04**±0.79 | 31.70±1.55 | **32.72**±0.18 |

**More Results** In Appendix F, we provide more experimental results. We report ensemble test accuracy of the baselines and FedGO, demonstrating a larger improvement compared to test accuracy. We also provide results for cases where the server dataset is different from the client datasets, as well as for data-free approaches when no server dataset is available, showing significant performance gains over the baselines. Additionally, we report the performance of FedGO with a reduced server dataset and various discriminator training epochs, showing that even with only 20% of the server dataset, FedGO achieves a performance gain of 15%p over FedAVG. Furthermore, FedGO outperforms the baselines even with significantly fewer discriminator training epochs.

In Appendix G, a comprehensive analysis of communication costs, privacy, and computational costs for FedGO and baselines is provided.

## 5 CONCLUSION

We proposed the FedGO algorithm, which effectively addresses the challenge of client data heterogeneity. Our algorithm was proposed based on theoretical analysis of optimal ensemble distillation, and various experimental results demonstrated its high performance and fast convergence rate under various scenarios with and without extra server dataset. Due to page limit, limitation and broader impact of our work are provided in Appendices H and I, respectively.

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

## A  FORMAL STATEMENT AND PROOF OF THEOREM 2

In addition to the setups and definitions introduced in Section 2, we assume binary classification task, i.e., $y(x) \in [0,1]$ and $h(x) \in \{0,1\}$, coupled with $\ell_1$ loss.

We first present some definitions and a lemma.

**Definition A.1.** (Kifer et al., 2004, Definition 1) For two distributions $q$ and $q'$ over a domain $\mathcal{X}$, let $\mathcal{H}$ denote a hypothesis class on $\mathcal{X}$ and $I(h)$ for $h \in \mathcal{H}$ denote the set $\{x \in \mathcal{X} : h(x) = 1\}$. The $\mathcal{H}$-divergence between $q$ and $q'$ is

$$d_{\mathcal{H}}(q, q') = 2 \sup_{h \in \mathcal{H}} |Pr_{x \sim q}[I(h)] - Pr_{x \sim q'}[I(h)]|. \tag{12}$$

**Definition A.2.** For a hypothesis space $\mathcal{H}$, the symmetric difference hypothesis space $\mathcal{H} \triangle \mathcal{H}$ is the set of hypotheses

$$g \in \mathcal{H} \triangle \mathcal{H} \Leftrightarrow g(x) = h(x) \oplus h'(x) \text{ for some } h, h' \in \mathcal{H} \tag{13}$$

where $\oplus$ is the XOR function.

**Lemma A.1.** For hypotheses $h, h' \in \mathcal{H}$ and distributions $q, q'$ on $\mathcal{X}$, we have

$$|\mathcal{L}_q(h, h') - \mathcal{L}_{q'}(h, h')| \leq \frac{1}{2} d_{\mathcal{H} \triangle \mathcal{H}}(q, q'). \tag{14}$$

*Proof.* By the definition of $\mathcal{H} \triangle \mathcal{H}$-distance, we have

$$d_{\mathcal{H} \triangle \mathcal{H}}(q, q') = 2 \sup_{h \in \mathcal{H}} |Pr_{x \sim q}[h(x) \neq h'(x)] - Pr_{x \sim q'}[h(x) \neq h'(x)]| \tag{15}$$

$$= 2 \sup_{h \in \mathcal{H}} |\mathcal{L}_q(h, h') - \mathcal{L}_{q'}(h, h')| \tag{16}$$

$$\geq 2|\mathcal{L}_q(h, h') - \mathcal{L}_{q'}(h, h')|, \tag{17}$$

which completes the proof.  □

Now we are ready to present the formal statement and proof of Theorem 2.

**Theorem A.1.** For binary classification task with $\ell_1$ loss, consider hypothesis class $\mathcal{H}$ such that $h \in \mathcal{H}$ outputs 0 or 1 and its spanned hypothesis class $\bar{\mathcal{H}} \triangleq \{\sum_{k=1}^{K} w_k \cdot h_k | h_k \in \mathcal{H}, w_k : \mathcal{X} \to [0,1]$ for all $k = 1, ..., K, \sum_{k=1}^{K} w_k = 1\}$. For any $h \in \mathcal{H}$ and $(\sum_{k=1}^{K} w_k \cdot h_k) \in \bar{\mathcal{H}}$, the following holds:

$$\mathcal{L}_p(h) \leq \mathcal{L}_p(\sum_{k=1}^{K} w_k \cdot h_k) + \mathcal{L}_{p_s}(h, \sum_{k=1}^{K} w_k \cdot h_k) + \frac{1}{2} d_{\bar{\mathcal{H}} \triangle \bar{\mathcal{H}}}(p, p_s). \tag{18}$$

*Proof.* We have

$$\mathcal{L}_p(h) = \mathbf{E}_p[l(h(x), y(x)] \tag{19}$$

$$\leq \mathbf{E}_p[l(h(x), (\sum_{k=1}^{K} w_k \cdot h_k)(x)] + \mathbf{E}_p[l((\sum_{k=1}^{K} w_k \cdot h_k)(x), y(x)] \tag{20}$$

$$= \mathcal{L}_p(\sum_{k=1}^{K} w_k \cdot h_k) + \mathcal{L}_p(h, \sum_{k=1}^{K} w_k \cdot h_k) \tag{21}$$

by triangle inequality (Ben-David et al., 2006; Crammer et al., 2008).

Since $A \leq B + |A - B|$, letting $A = \mathcal{L}_p(h, \sum_{k=1}^{K} w_k \cdot h_k)$, $B = \mathcal{L}_{p_s}(h, \sum_{k=1}^{K} w_k \cdot h_k)$, the RHS of (21) is upper-bounded by

$$\mathcal{L}_p(\sum_{k=1}^{K} w_k \cdot h_k) + \mathcal{L}_{p_s}(h, \sum_{k=1}^{K} w_k \cdot h_k) + |\mathcal{L}_p(h, \sum_{k=1}^{K} w_k \cdot h_k) - \mathcal{L}_{p_s}(h, \sum_{k=1}^{K} w_k \cdot h_k)| \tag{22}$$

$$= \mathcal{L}_p(\sum_{k=1}^{K} w_k \cdot h_k) + \mathcal{L}_{p_s}(h, \sum_{k=1}^{K} w_k \cdot h_k) + \frac{1}{2} d_{\bar{\mathcal{H}} \triangle \bar{\mathcal{H}}}(p, p_s) \tag{23}$$

by the definition of $d_{\bar{\mathcal{H}} \triangle \bar{\mathcal{H}}}$ and Lemma A.1. Thus, we prove Theorem A.1.  □

## B  PROOF OF THEOREM 3

Before we start the proof of Theorem 3, we present the following lemma with the setups and definitions introduced in Section 2.

**Lemma B.1.** Let the loss function $l$ be convex and $\{h_k\}_{k=1}^K \subset \mathcal{H}$. For the weight functions $\{w_k^*\}_{k=1}^K$ defined in Theorem 3, the following holds:

$$\mathcal{L}_p(\textstyle\sum_k w_k^* \cdot h_k) \leq \textstyle\sum_k \pi_k \cdot \mathcal{L}_{p_k}(h_k). \tag{24}$$

*Proof.* Note that

$$\mathcal{L}_p(\textstyle\sum_k w_k^* \cdot h_k) = \mathbb{E}_{x \sim p}\left[l\left(\textstyle\sum_k w_k^*(x) \cdot h_k(x), y(x)\right)\right] \tag{25}$$

$$= \int l\left(\textstyle\sum_k w_k^*(x) \cdot h_k(x), y(x)\right) \cdot p(x)dx \tag{26}$$

$$= \int l\left(\textstyle\sum_k w_k^*(x) \cdot h_k(x), y(x)\right) \cdot \textstyle\sum_j \pi_j \cdot p_j(x)dx \tag{27}$$

$$= \int l\left(\textstyle\sum_k w_k^*(x) \cdot h_k(x), \textstyle\sum_k w_k^*(x) \cdot y(x)\right) \cdot \textstyle\sum_j \pi_j \cdot p_j(x)dx \tag{28}$$

$$\leq \int \left(\textstyle\sum_k w_k^*(x) \cdot l\left(h_k(x), y(x)\right)\right) \cdot \textstyle\sum_j \pi_j \cdot p_j(x)dx \tag{29}$$

$$= \int \textstyle\sum_k \frac{\pi_k(x) \cdot p_k(x)}{\sum_{i=1}^K \pi_i \cdot p_i(x)} \cdot l\left(h_k(x), y(x)\right) \cdot \textstyle\sum_j \pi_j \cdot p_j(x)dx \tag{30}$$

$$= \sum_k \int \pi_k \cdot p_k(x) \cdot l\left(h_k(x), y(x)\right) dx \tag{31}$$

$$= \sum_k \pi_k \cdot \int l\left(h_k(x), y(x)\right) \cdot p_k(x)dx \tag{32}$$

$$= \sum_k \pi_k \cdot \mathcal{L}_{p_k}(h_k), \tag{33}$$

where (29) holds due to the convexity of loss function $l(\cdot, \cdot)$. This completes the proof. $\square$

Now we present the proof of Theorem 3.

*Proof.* For $h \in \mathcal{H}$, we have

$$\mathcal{L}_p(h) = \mathbb{E}_{x \sim p}\left[l\left(h(x), y(x)\right)\right] \tag{34}$$

$$= \int l\left(h(x), y(x)\right) \cdot p(x)dx \tag{35}$$

$$= \int l\left(h(x), y(x)\right) \cdot \textstyle\sum_k \pi_k \cdot p_k(x)dx \tag{36}$$

$$= \textstyle\sum_k \pi_k \cdot \int \left[l\left(h(x), y(x)\right)\right] \cdot p_k(x)dx \tag{37}$$

$$= \textstyle\sum_k \pi_k \cdot \mathcal{L}_{p_k}(h) \tag{38}$$

$$\geq \textstyle\sum_k \pi_k \cdot \mathcal{L}_{p_k}(h_{p_k}^*). \tag{39}$$

Hence, it suffices to show that

$$\mathcal{L}_p(\textstyle\sum_k w_k^* \cdot h_{p_k}^*) \leq \textstyle\sum_k \pi_k \cdot \mathcal{L}_{p_k}(h_{p_k}^*), \tag{40}$$

and this is the direct result of Lemma B.1 with $\{h_k\}_{k=1}^K = \{h_{p_k}^*\}_{k=1}^K$. $\square$

## C  GENERALIZATION BOUND WITH EMPIRICAL LOSS MINIMIZER

In this section, we present the generalization loss bound of the ensemble of empirical loss minimizers of clients with our weighting method.

**Theorem C.1.** For binary classification task with $\ell_1$ loss, the following holds for our weighting function $\{w_k^*\}_{k=1}^K$ defined in Theorem 3:

$$\mathcal{L}_p(\sum_{k=1}^K w_k^* \cdot h_{\hat{p}_k}^*) \leq \sum_{k=1}^K \pi_k \cdot \left[ \mathcal{L}_{\hat{p}_k}(h_{\hat{p}_k}^*) + \frac{4 + \sqrt{\log(\tau_{\mathcal{H}}(2n_k))}}{(\delta/K)\sqrt{2n_k}} \right] \tag{41}$$

$$\leq \mathcal{L}_{\hat{p}}(h_{\hat{p}}^*) + \sum_{k=1}^K \pi_k \cdot \frac{4 + \sqrt{\log(\tau_{\mathcal{H}}(2n_k))}}{(\delta/K) \cdot \sqrt{2n_k}}, \tag{42}$$

where $\hat{p}_k$ is the empirical distribution by sampling $n_k$ data points i.i.d. according to $p_k$, $\hat{p} = \sum_{k=1}^K \pi_k \cdot \hat{p}_k$, and $\tau_{\mathcal{H}}$ is growth function bounded by polynomial of the VC-dimension of $\mathcal{H}$.

Compared to (4), we can see that the ensemble of empirical loss minimizers with our weighting method has a tighter generalization bound without the factor of $(K + 1)$.

Before we prove Theorem C.1, we present the following theorem.

**Theorem C.2.** (Shalev-Shwartz & Ben-David, Theorem 6.11) Let $\mathcal{H}$ be a hypothesiss class and let $\tau_{\mathcal{H}}$ be its growth function. Then, for every distribution $q$ on $\mathcal{X}$ and every $\delta \in (0, 1)$, with probability of at least $1 - \delta$ over the $m$ i.i.d. choice of $S \sim q^m$ with its empirical distribution $\hat{q}$, we have

$$|\mathcal{L}_q(h) - \mathcal{L}_{\hat{q}}(h)| \leq \frac{4 + \sqrt{\log(\tau_{\mathcal{H}}(2m))}}{\delta\sqrt{2m}}. \tag{43}$$

We also present the bound of growth function $\tau_{\mathcal{H}}$.

**Lemma C.1.** (Shalev-Shwartz & Ben-David, Lemma 6.10) Let $\mathcal{H}$ be a hypothesis class with VC-dimension of $\mathcal{H}$ is smaller than $d$, i.e. $VCDim(\mathcal{H}) \leq d < \infty$. Then, for all $m$, $\tau_{\mathcal{H}}(m) \leq \sum_{i=0}^d \binom{m}{i}$. In particular, if $m > d + 1$, then $\tau_{\mathcal{H}}(m) \leq (em/d)^d$, where $e$ is Euler's number.

Now we present the proof of Theorem C.1.

*Proof.* By the result of Lemma B.1 with $\{h_k\}_{k=1}^K = \{h_{\hat{p}_k}^*\}_{k=1}^K$, we can derive

$$\mathcal{L}_p(\sum_k w_k^* \cdot h_{\hat{p}_k}^*) \leq \sum_k \pi_k \cdot \mathcal{L}_{p_k}(h_{\hat{p}_k}^*). \tag{44}$$

Also we note that $S_k \sim p_k^{n_k}$. We can derive the following inequality for $k = 1, ..., K$ using Theorem C.2. With probability at least of $1 - (\delta/K)$,

$$\mathcal{L}_{p_k}(h_{\hat{p}_k}^*) \leq \mathcal{L}_{\hat{p}_k}(h_{\hat{p}_k}^*) + \frac{4 + \sqrt{\log(\tau_{\mathcal{H}}(2n_k))}}{(\delta/K)\sqrt{2n_k}}. \tag{45}$$

where $\tau_{\mathcal{H}}$ is growth function bounded by polynomial of the VC-dimension of $\mathcal{H}$.

By the union bound, we have

$$P\left[ \bigcap_{k=1}^K \left( \mathcal{L}_{p_k}(h_{\hat{p}_k}^*) \leq \mathcal{L}_{\hat{p}_k}(h_{\hat{p}_k}^*) + \frac{4 + \sqrt{\log(\tau_{\mathcal{H}}(2n_k))}}{(\delta/K)\sqrt{2n_k}} \right) \right] \tag{46}$$

$$= 1 - P\left[ \bigcup_{k=1}^K \left( \mathcal{L}_{p_k}(h_{\hat{p}_k}^*) \geq \mathcal{L}_{\hat{p}_k}(h_{\hat{p}_k}^*) + \frac{4 + \sqrt{\log(\tau_{\mathcal{H}}(2n_k))}}{(\delta/K)\sqrt{2n_k}} \right) \right] \tag{47}$$

$$\geq 1 - \sum_{k=1}^K P\left[ \left( \mathcal{L}_{p_k}(h_{\hat{p}_k}^*) \geq \mathcal{L}_{\hat{p}_k}(h_{\hat{p}_k}^*) + \frac{4 + \sqrt{\log(\tau_{\mathcal{H}}(2n_k))}}{(\delta/K)\sqrt{2n_k}} \right) \right] \tag{48}$$

$$\geq 1 - \sum_{k=1}^K (\delta/K) \tag{49}$$

$$\geq 1 - \delta. \tag{50}$$

Hence, with probability at least $1 - \delta$, following inequality holds for all $k = 1, .., K$:

$$\mathcal{L}_{p_k}(h_{\hat{p}_k}^*) \leq \mathcal{L}_{\hat{p_k}}(h_{\hat{p}_k}^*) + \frac{4 + \sqrt{\log(\tau_{\mathcal{H}}(2n_k))}}{(\delta/K)\sqrt{2n_k}}. \tag{51}$$

Furthermore, by definition of $\hat{p}$,

$$\mathcal{L}_{\hat{p}}(h_{\hat{p}}^*) = \sum_k \pi_k \cdot \mathcal{L}_{\hat{p_k}}(h_{\hat{p}}^*) \tag{52}$$

$$\geq \sum_k \pi_k \cdot \mathcal{L}_{\hat{p_k}}(h_{\hat{p}_k}^*). \tag{53}$$

By combining the above results, with probability of at least $1 - \delta$, we have

$$\mathcal{L}_p(\sum_{k=1}^K w_k^* \cdot h_{\hat{p}_k}^*) \leq \sum_k \pi_k \cdot \mathcal{L}_{p_k}(h_{\hat{p}_k}^*) \tag{54}$$

$$\leq \sum_{k=1}^K \pi_k \cdot \left[ \mathcal{L}_{\hat{p_k}}(h_{\hat{p}_k}^*) + \frac{4 + \sqrt{\log(\tau_{\mathcal{H}}(2n_k))}}{(\delta/K)\sqrt{2n_k}} \right] \tag{55}$$

$$\leq \sum_{k=1}^K \left[ \pi_k \cdot \mathcal{L}_{\hat{p_k}}(h_{\hat{p}_k}^*) + \pi_k \cdot \frac{4 + \sqrt{\log(\tau_{\mathcal{H}}(2n_k))}}{(\delta/K)\sqrt{2n_k}} \right] \tag{56}$$

$$\leq \mathcal{L}_{\hat{p}}(h_{\hat{p}}^*) + \sum_{k=1}^K \pi_k \cdot \frac{4 + \sqrt{\log(\tau_{\mathcal{H}}(2n_k))}}{(\delta/K)\sqrt{2n_k}}. \tag{57}$$

This completes the proof. $\qquad\qquad\qquad\qquad\qquad\qquad\qquad\qquad\qquad\qquad\qquad\square$

# D  DESCRIPTION OF FEDGO

Figure 4 illustrates the operation of FedGO. Algorithm 2 presents a pseudo-code of FedGO. For training discriminators, each client optimizes the GAN loss with respect to its labeled dataset. A pseudo-code of client discriminator update is provided in Algorithm 3.

---

**Algorithm 2** FedGO algorithm with $K$ clients for $T$ communication rounds. $f(\cdot; \theta)$ stands for the model with parameter $\theta$, $\mu$ stands for the step size, and $\Phi_k(x)$ stands for the odds value of $D_k(x)$.

---

**Require:** Client labeled dataset $\{S_k\}_{k=1}^K$
1: Initialize server model $f(\cdot, \theta_s^0)$ with parameter $\theta_s^0$
2: Prepare generate $G$ and unlabeled dataset $U$ $\qquad\qquad\qquad$ ▷ By one of the methods in Table 1
3: **parfor** client $k \in \{1, 2, ..., K\}$ **do**
4: $\qquad D_k \leftarrow DiscriminatorUpdate(G, S_k)$ $\qquad\qquad\qquad$ ▷ Detailed in Algorithm 3
5: **end parfor**
6: **for** communication round $t = 1$ to $T$ **do**
7: $\qquad A^t \leftarrow$ sample $\lfloor C \cdot K \rfloor$ clients
8: $\qquad$ **parfor** client $k \in A^t$ **do**
9: $\qquad\qquad \theta_k^t \leftarrow ClientUpdate(\theta_s^{t-1}, S_k)$ $\qquad\qquad\qquad$ ▷ Gradient update $\theta_s^{t-1}$ with $S_k$
10: $\qquad$ **end parfor**
11: $\qquad \theta_s^t \leftarrow \sum_{k \in A^t} \frac{n_k}{\sum_{i \in A^t} n_i} \cdot \theta_k^t$
12: $\qquad$ **for** server train epoch $e = 1$ to $E_s$ **do**
13: $\qquad\qquad$ **for** unlabeled minibatch $u \in U$ **do**
14: $\qquad\qquad\qquad \tilde{y}(u) \leftarrow \sigma(\sum_{k \in A^t} w_k^*(u) \cdot f(u; \theta_k^t))$ $\qquad\qquad$ ▷ $w_k^*(u) = \frac{n_k \cdot \Phi_k(u)}{\sum_{i \in A^t} n_i \cdot \Phi_i(u)}$
15: $\qquad\qquad\qquad \theta_s^t \leftarrow \theta_s^t - \mu \cdot \nabla_{\theta_s^t} \mathrm{KL}(\tilde{y}(u), \sigma(f(u; \theta_s^t)))$
16: $\qquad\qquad$ **end for**
17: $\qquad$ **end for**
18: **end for**
19: **return** $f(\cdot, \theta_s^T)$

---

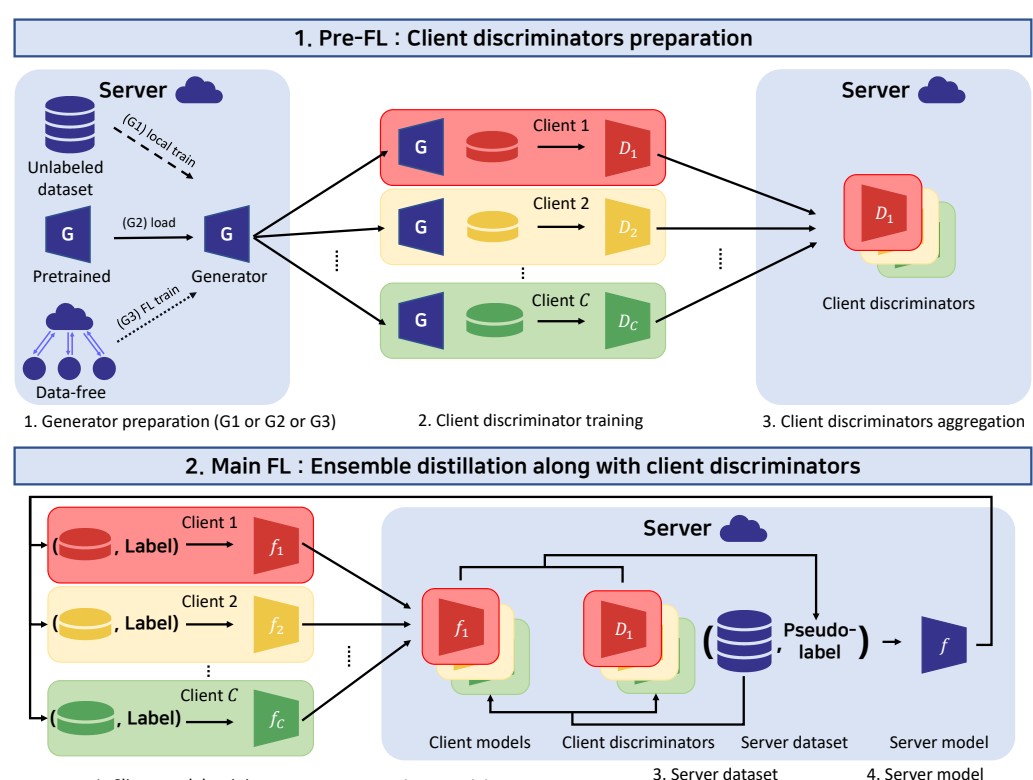

Figure 4: Illustration of our FedGO algorithm.

---

**Algorithm 3** Discriminator update for $E_d$ epochs. $\mu_d$ stands for the step size and $D(\cdot, \theta)$ is the parameterized discriminator with parameter $\theta$.

---

**Require:** Generator $G$, labeled dataset $S$
1: Initialize discriminator model $D(\cdot, \theta_d^0)$ with parameter $\theta_d^0$
2: **for** epoch $e = 1$ to $E_d$ **do**
3:     $\theta_d^e \leftarrow \theta_d^{e-1}$
4:     **for** minibatch $m \in S$ **do**
5:         $(x_{real}, y) \leftarrow$ (images, labels) pair of minimatch $m$
6:         $x_{fake} \leftarrow$ generated images by generator $G$
7:         $Loss_{GAN}(D(\cdot, \theta_d^e)) \leftarrow \log(D(x_{real}, \theta_d^e) + \log(1 - D(x_{fake}, \theta_d^e))$
8:         $\theta_d^e \leftarrow \theta_d^e - \mu_d \nabla_{\theta_d^e} Loss_{GAN}(D(\cdot, \theta_d^e))$      ▷ Update with gradient for vanilla GAN loss
9:     **end for**
10: **end for**
11: **return** $D(\cdot, \theta_d^{E_d}))$

---

# E EXPERIMENTAL DETAILS

All experiments were conducted in Python 3.8.12 environment using a 64-core Intel 2.90GHz Xeon Gold 6226R CPU with 512GB memory, and an RTX 3090 GPU. We also implemented the algorithms using PyTorch with version 1.11.0.

## E.1 DETAILED EXPERIMENTAL SETTING AND ANALYSIS OF TOY EXAMPLE (FIGURE 1)

For the toy example in Figure 1, the dataset is generated from a mixture of four Gaussian distributions, each with a variance of 3. The top row of Figure 5 shows the global data distribution and the datasets held by four clients. Each point represents data, with its color indicating the class label:

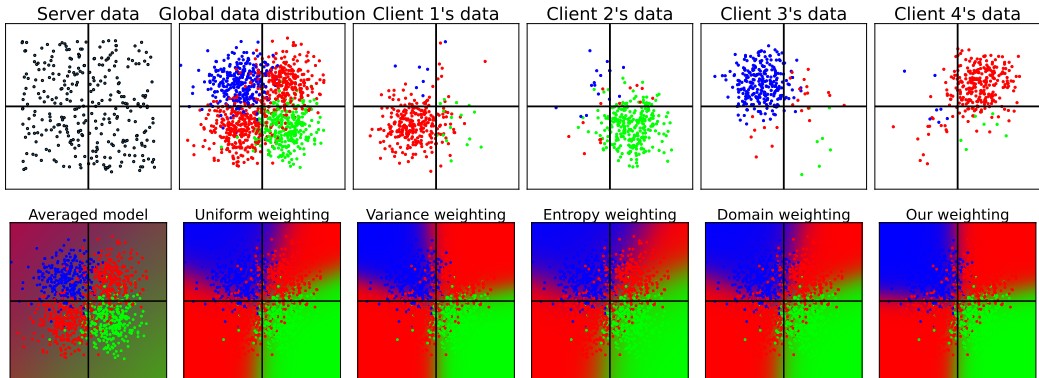

Figure 5: Top row represents the server and clients' datasets. Bottom row, showing the decision boundaries of the aggregated models, is the same as Figure 1 and copied here for ease of analysis.

data from Gaussians with means at (4, 4) and (-4, -4) are labeled as Red, data from the Gaussian with mean at (-4, 4) as Blue, and data from the Gaussian with mean at (4, -4) as Green. Each Gaussian provides 300 data samples. Each client holds 90% of data from the Gaussian whose mean is in a certain quadrant (the 3rd, 4th, 2nd, 1st quadrants for Clients 1, 2, 3, and 4, respectively), and the remaining 10% from Gaussians with means in the other quadrants. The clients' global dataset comprises 1200 samples, with 300 from each Gaussian. The server unlabeled dataset comprises 300 data, uniformly distributed on the square $[-12, 12] \times [-12, 12]$.

Each client trains a 3-layer MLP classifier for 2 epochs using its dataset, and a 3-layer discriminator for 1 epoch using its dataset as real dataset and server dataset as fake dataset. We used Adam (Kingma & Ba, 2014) with learning rate 0.001 and $(\beta_1, \beta_2) = (0.9, 0.999)$ for classifier optimizer, and RMSprop (Hinton et al.) with learning rate 0.00005 for discriminator optimizer. Also we used a batch size of 64 for both.

The bottom row of Figure 5 (same as Figure 1) illustrates the decision boundaries of server models. The leftmost plot is from the model with averaged client model parameters, while the remaining plots are from the server models trained via ensemble distillation for 2 epochs using pseudo-labeled dataset: the global dataset is pseudo-labeled using uniform weighting (Lin et al., 2020), variance weighting (Cho et al., 2022), entropy weighting (Deng et al., 2023; Park et al., 2024), domain-aware weighting (Wang et al., 2023a) , and our weighting method. The background color indicates the decision boundary in RGB channels. Given the Gaussian distributions, the optimal decision rule is red in the 1st and 3rd quadrants, blue in the 2nd quadrant, and green in the 4th quadrant. Thus, the oracle decision boundary aligns with the x-axis and y-axis, depicted by black lines.

The averaged parameter model exhibits a blurred decision boundary compared to models trained via ensemble distillation. Furthermore, among the models with ensemble distillation, the decision boundary of the model trained via our weighting method is closest to the oracle decision boundary.

### E.2 DETAILED EXPERIMENTAL SETTINGS FOR IMAGE CLASSIFICATION TASKS

**Hyperparameter Tuning** We identified the best-performing hyperparameters on CIFAR-100 with Dirichlet $\alpha = 0.05$ and used the same values for other settings. During the ensemble distillation process, we trained both clients and server with the Adam optimizer (Kingma & Ba, 2014) at a learning rate of 0.001 with batch size 64, without weight decay. The $(\beta_1, \beta_2)$ parameters for Adam were set to (0.9, 0.999). Additionally, we applied cosine annealing (Loshchilov & Hutter, 2016) to decay the server learning rate until the final communication round $T = 100$ as in Lin et al. (2020), except for the results of F.2 and F.4.

For the client and server classifier training epochs, we performed a grid search to find the optimal number of training epochs. The initial grid was {5, 10, 30, 50}, and the experiments were conducted with 30 client epochs and 10 server epochs ($E_s = 10$) for CIFAR-10/100. To leverage the increased

number of steps due to the additional number of data, experiments on ImageNet100 were conducted with 10 client classifier epochs and 3 server classifier epochs ($E_s = 3$).

To train the generator utilized by our FedGO from scratch, we trained the WGAN-GP model following the training method proposed in Gulrajani et al. (2017). The generator and discriminator of WGAN-GP were trained using the Adam optimizer with a learning rate of 0.0002 and $(\beta_1, \beta_2) = (0, 0.9)$. The training was conducted with a batch size of 64 until the generator completed 100,000 gradient steps. The generator was updated every 5 steps of the discriminator, and a gradient penalty coefficient $\lambda$ of 10 was used.

When training a generator in a data-free setting, i.e., the case (G3), we applied the same hyperparameters as in the scratch training, except for the gradient steps. For gradient steps, we used the same number of training epochs for the local generator as those used in classifier training.

For the client discriminator, we adopted the hyperparameters from https://github.com/Ksuryateja/DCGAN-MNIST-pytorch/blob/master/gan_mnist.py and trained it with a batch size of 64 for 30 epochs for CIFAR-10/100, and for 10 epochs for ImageNet100. The optimizer Adam was used with a learning rate of 0.0002, and $(\beta_1, \beta_2) = (0.5, 0.999)$.

FedProx (Li et al., 2020) introduces a proximal term to the client training loss, which helps to address heterogeneity by penalizing large deviations from the server model. The proximal term is multiplied by a coefficient $\mu$ and added to the primary objective loss. We performed a grid search to tune the value of $\mu$ from $\{0.1, 0.05, 0.01, 0.005, 0.001, 0.0005, 0.0001\}$, and chose the best value $\mu = 0.001$.

FedHKT (Deng et al., 2023) and FedDS (Park et al., 2024) introduce a temperature parameter $\tau > 0$, which allows client weights to approach uniform weighting as $\tau$ increases. By following Deng et al. (2023), we set $\tau = 1$.

FedGKD (Yao et al., 2021) introduces an additional buffer of length $M$ on the server, where the server model is stored after each round. The server then creates an additional model with averaged parameters from the models stored in the buffer and sends this model to the clients each round. Each client uses a temperature parameter $\tau$ to compute the knowledge distillation loss on the received additional model, multiplies this loss by $\gamma/2$, and adds it to the primary objective loss. Consequently, it is necessary to tune three additional hyperparameters: $M$, $\tau$, and $\gamma$. We conducted a grid search with $M$ and $\tau$ in $\{1, 3, 5, 10\}$ and $\gamma$ in $\{0.1, 0.05, 0.01, 0.005, 0.001\}$. The best performing parameters were $M = 5$, $\tau = 3$, and $\gamma = 0.001$.

Similar to our FedGO, DaFKD (Wang et al., 2023a) utilizes discriminators to implement client weighting function. However, unlike FedGO, DaFKD trains the generator and discriminators collaboratively. To focus on the weighting method, the domain-aware weighting method in Figure 2 is implemented by only modifying the weighting step in our FedGO algorithm.

**Model Implementation** We used ResNet-18 (He et al., 2016) as the classification model, following the implementation from https://github.com/kuangliu/pytorch-cifar/blob/master/models/resnet.py. Additionally, our FedGO requires extra generator and discriminator models. When training the generator from scratch, we utilized the WGAN-GP model as proposed in Gulrajani et al. (2017), following its official open-source implementation[1]. We re-implemented this code in PyTorch for our experiments. For a pretrained off-the-shelf generator, we utilized StyleGAN-XL (Sauer et al., 2022) model pretrained on ImageNet (Krizhevsky et al., 2012) with resolution of $32{\times}32$. We downloaded the model parameters from https://github.com/autonomousvision/stylegan-xl and implemented the model using these parameters. For the client discriminator, we adopted a simple 4-layer CNN discriminator, following the implementation from https://github.com/Ksuryateja/DCGAN-MNIST-pytorch/blob/master/gan_mnist.py. To address the widely known overfitting issue of the discriminator (Adlam et al., 2019; Yang et al., 2022) and the resulting dominance of client weights, we employed a composition of two sigmoid activations for the discriminator output. This ensures that the odds value $\Phi_k$ for client $k$'s discriminator $D_k$ is constrained between 1 and $e$.

**Heterogeneous Client Data Split** To introduce non-iid distributions among client datasets, we ensured that each client's distribution follows a Dirichlet distribution Dir($\alpha$), similar as in Lin et al.

---

[1] https://github.com/igul222/improved_wgan_training

(2020); Wang et al. (2020a); Marfoq et al. (2022); Li et al. (2023). As the parameter $\alpha$ increases, each client tends to have a more homogeneous distribution, whereas smaller $\alpha$ values result in increased data heterogeneity among clients. We conducted experiments for each dataset with $\alpha$ values of 0.1 and 0.05. The number of data samples that each client has per class for CIFAR-10/100 datasets with $\alpha$ values of 0.1 and 0.05 is illustrated in Figures 6 and 7. It's worth noting that ImageNet100 also has 100 classes, so the trends observed in CIFAR-100 would likely align with those in ImageNet100. We can observe that when $\alpha = 0.05$, the difference in the number of data samples per class for each client is more pronounced compared to when $\alpha = 0.1$. This results in more skewed distributions for individual clients.

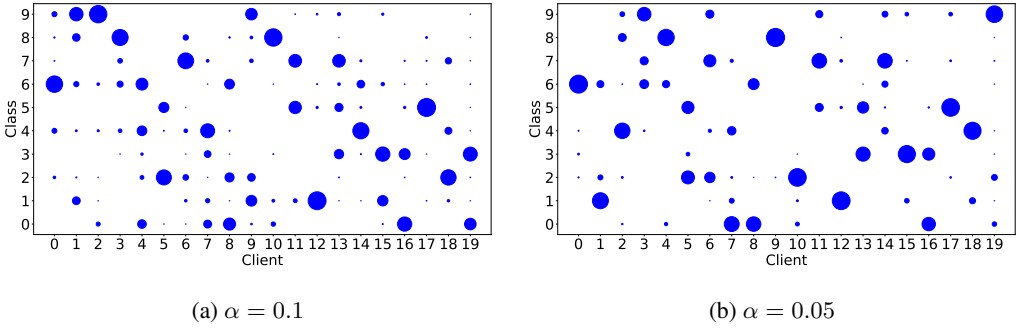

(a) $\alpha = 0.1$         (b) $\alpha = 0.05$

Figure 6: Client data split for CIFAR-10 with $\alpha = 0.1, 0.05$.

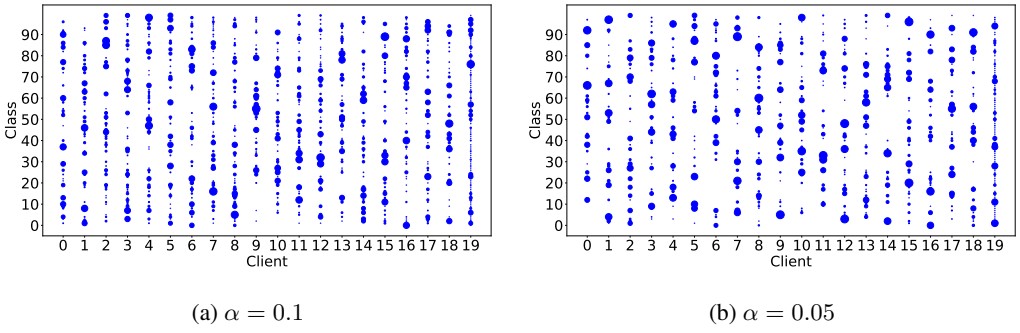

(a) $\alpha = 0.1$         (b) $\alpha = 0.05$

Figure 7: Client data split for CIFAR-100 with $\alpha = 0.1, 0.05$.

**Details for Dataset** We normalized the pixel values of all image datasets to fall within the range $[-1, 1]$, ensuring that the generated data also has pixel values within this range. Additionally, for both the training datasets of clients and the server's unlabeled dataset, we conducted further data augmentation using PyTorch's random horizontal flip.

**Selection of Acc$_{target}$** We used the highest multiple of 5 of the test accuracy (%) achieved by the FedAVG algorithm within 100 rounds for all five different random seeds as Acc$_{target}$ for Table 3.

# F ADDITIONAL EXPERIMENTAL RESULTS

## F.1 ENSEMBLE TEST ACCURACY COMPARISON AND ANALYSIS

Figure 8 shows the ensemble test accuracy on the server's unlabeled dataset during the training process for our FedGO algorithm and the baseline ensemble algorithms: FedDF, FedGKD$^+$, and DaFKD. It demonstrates that using pseudo-labels generated by theoretically guaranteed weighting methods allows the server to achieve higher final performance and faster convergence.

However, in Table 2 of the paper, the performance gap between our method and the baselines on CIFAR-100 and ImageNet100 was not as large as that on CIFAR-10. We infer the reason from

Theorem 2. The second term on the RHS of Theorem 2 can be interpreted as the distillation loss due to the difference between the hypothesis class and the spanned hypothesis class. Even if our ensemble is close to optimal, the knowledge-distilled server model may not follow the performance of the ensemble if it hard for a single model to learn the pseudo-labels, and we conjectures that it becomes harder as the number of classes increases.

To support our hypothesis, we show the minimum of mean distillation loss for five different random seeds during 100 round of communication rounds in Table 5. The distillation loss increases progressively from CIFAR-10 to CIFAR-100 to ImageNet100. In addition, the distillation loss is higher for $\alpha = 0.05$ than for $\alpha = 0.1$, which explains why the gap from the central training is larger for $\alpha = 0.05$.

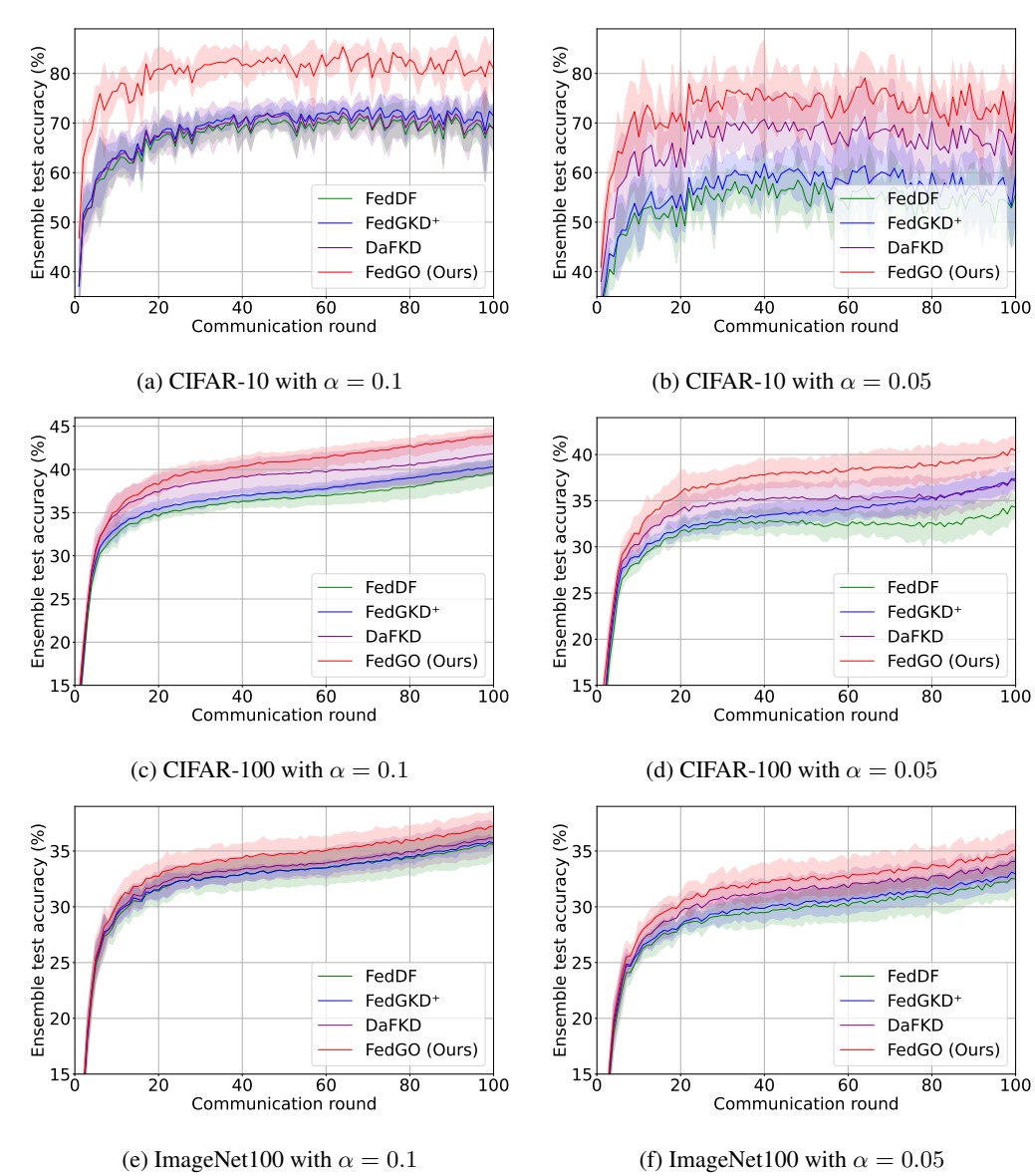

(a) CIFAR-10 with $\alpha = 0.1$

(b) CIFAR-10 with $\alpha = 0.05$

(c) CIFAR-100 with $\alpha = 0.1$

(d) CIFAR-100 with $\alpha = 0.05$

(e) ImageNet100 with $\alpha = 0.1$

(f) ImageNet100 with $\alpha = 0.05$

Figure 8: Ensemble test accuracy (%) of FedGO and baselines over communication rounds on three image datasets with $\alpha = 0.1, 0.05$.

Table 5: Minimum mean distillation loss of FedGO on three image datasets with $\alpha = 0.1, 0.05$.

| Dataset | CIFAR-10 | CIFAR-100 | ImageNet100 |
|---|---|---|---|
| $\alpha = 0.1$ | 0.175 | 0.237 | 0.363 |
| $\alpha = 0.05$ | 0.266 | 0.348 | 0.539 |

## F.2 ENSEMBLE DISTILLATION WITH A DIFFERENT SERVER DATASET

Our theoretical justification of constituting an optimal ensemble in Corollary 1 allows heterogeneity between the server data distribution $p_s$ and the client average distribution $p$. To demonstrate the effectiveness of FedGO when $p_s \neq p$ which makes more sense in practice, we report the results when clients have the half of the CIFAR-10 dataset and the server has the half of the CIFAR-100 (unlabeled) dataset, in Table 6. The experimental results demonstrate that ensemble distillation even with heterogeneous server dataset is helpful in improving the performance. Furthermore, by employing optimal model ensemble, our FedGO algorithm, with theoretical performance guarantee, shows improvement over FedDF and DaFKD.

Table 6: Server test accuracy (%) and ensemble test accuracy (%) of our FedGO and baselines with heterogeneous server dataset: CIFAR-10 for client dataset and CIFAR-100 for server's unlabeled dataset.

| | | FedAVG | FedDF | DaFKD | **FedGO (ours)** |
|---|---|---|---|---|---|
| $\alpha = 0.1$ | Server test accuracy | 58.65±5.75 | 59.89±1.88 | 60.84±2.65 | **60.92**±1.95 |
| | Ensemble test accuracy | - | 62.62±0.90 | 63.88 ± 2.02 | **64.23**±1.29 |
| $\alpha = 0.05$ | Server test accuracy | 46.61±8.54 | 49.21±4.48 | 52.31±4.26 | **52.89**±3.47 |
| | Ensemble test accuracy | - | 56.06±207 | 59.30 ±1.33 | **60.43** ±0.56 |

## F.3 RESULTS WITH ALTERNATIVE MODEL ARCHITECTURES

In the main paper, we conducted experiments with ResNet-18 model structure. In this subsection, we present the results with VGG11 (Simonyan & Zisserman, 2014) (with BatchNorm Layers (Ioffe & Szegedy, 2015)) and ResNet-50 models. For VGG11, both the client and server models are trained using SGD with a learning rate of 0.01 and momentum of 0.9, and all the other settings including hyperparameters are kept identical to those in the main paper. We implemented VGG11 based on https://github.com/chengyangfu/pytorch-vgg-cifar10. For ResNet-50, all the settings including optimizer and hyperparameters are set to the same as the main paper. Table 7 presents the server test accuracy of FedGO and baseline algorithms with the aforementioned model structures on CIFAR-10 with $\alpha = 0.1$ after 100 communication rounds.

Table 7: Server test accuracy (%) of central training, FedDF, FedGKD[+] and FedGO on CIFAR-10 with $\alpha = 0.1$ after 100 communication rounds, when utilizing VGG11 and ResNet-50.

| | VGG11 | ResNet-50 |
|---|---|---|
| Central training | $83.27 \pm 0.60$ | $85.12 \pm 0.44$ |
| FedDF | $68.59 \pm 4.65$ | $65.21 \pm 4.62$ |
| FedGKD[+] | $67.81 \pm 3.60$ | $66.21 \pm 3.01$ |
| **FedGO (ours)** | $72.53 \pm 4.10$ | $75.52 \pm 4.30$ |

We can see that our FedGO algorithm consistently achieves performance gains over FedDF and FedGKD[+] across different model structures.

### F.4 DATA-FREE FEDGO

In practice, the server may have no extra dataset. In this case, we first prepare a generator and then generate a distillation dataset using the generator. The generator can either be an off-the-shelf pretrained model or trained through an FL approach (Rasouli et al., 2020; Guerraoui et al., 2020; Li et al., 2022; Wang et al., 2023c; Fan & Liu, 2020; Behera et al., 2022; Hardy et al., 2019; Xiong et al., 2023; Zhang et al., 2021; 2023a), corresponding to the 3rd and 4th scenarios in Table 1 in our main paper, respectively.

Figures 9 and 10 present the results for the two data-free approaches with 100 clients on CIFAR-10 dataset. We employed styleGAN (Karras et al., 2019) pretrained with ImageNet dataset for the off-the-shelf generator, and applied the FedGAN algorithm (Rasouli et al., 2020) for training a generator. For both the approaches, our FedGO shows performance gains in server test accuracy, ensemble test accuracy, and ensemble test loss compared to the uniform weighting of FedDF (Lin et al., 2020) and the domain-aware weighting of DaFKD (Wang et al., 2023a). In particular, the improvement of FedGO over FedDF is much larger than that of DaFKD over FedDF. Note that the ensemble test loss of DaFKD becomes larger than that of FedDF after a certain round.

In both data-free approaches, we have $p_s = p_g$, under which our weighting method is optimal for $\forall x \in p_s = p_g$ from Theorem 4 in our main paper. Note that the distance between $p$ and $p_g = p_s$ for the generator trained from FedGAN is smaller than that for the off-the-shelf generator. Consequently, despite using a simpler generator trained on a smaller dataset, we observe that the performance of FedGO using the generator trained from FedGAN is slightly better than that using the off-the-shelf generator.

Finally, we can observe performance degradation compared to the case where the distillation is performed on a real dataset. This can be attributed to the naive reuse of generated images, which has been identified as a cause of performance degradation (Yoon et al., 2024; Wang et al., 2024). An interesting future work would be on improving the performance of knowledge distillation using generated images. Still, experimental results demonstrate that ensemble distillation is beneficial in improving performance even with generated images. Furthermore, by employing an optimal model ensemble, our FedGO shows improvement over FedDF and DaFKD.

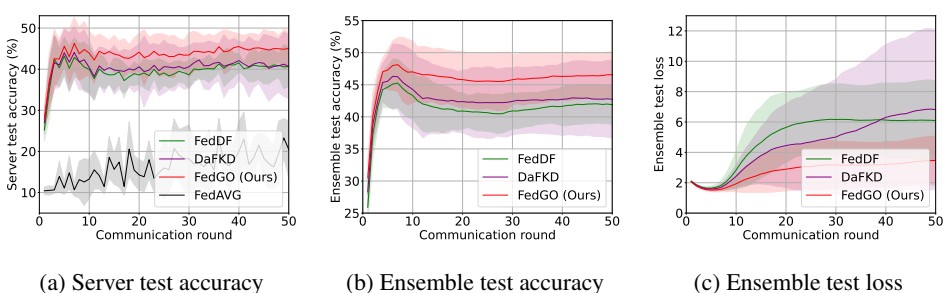

|     (a) Server test accuracy     |     (b) Ensemble test accuracy     |     (c) Ensemble test loss     |

Figure 9: Test accuracy of server model (%), ensemble test accuracy (%), and test loss of ensemble model of the data-free FedGO with an off-the-shelf generator (the case (G2)+(D2) of Table 1) and baselines with 100 clients on the CIFAR-10 dataset with $\alpha = 0.05$.

### F.5 IMPACT OF SERVER MODEL HYPERPARAMETERS ON PERFORMANCE

#### F.5.1 AMOUNT OF UNLABELED DATA

Figure 11 shows the test accuracy of the server model and the test accuracy of the ensemble model during the training process for our FedGO algorithm. We conducted experiments by reducing the server dataset size to 50% and 20% of the size assumed in our main CIFAR-10 experiments. For these experiments, the server epochs were adjusted to ensure the same number of gradient steps: doubled for 50% and quintupled for 20%, while keeping other hyperparameters the same.

Figure 11 demonstrates that when the server dataset size decreases, the test accuracy of the ensemble model remains nearly consistent, while that of the server model decreases. This suggests that

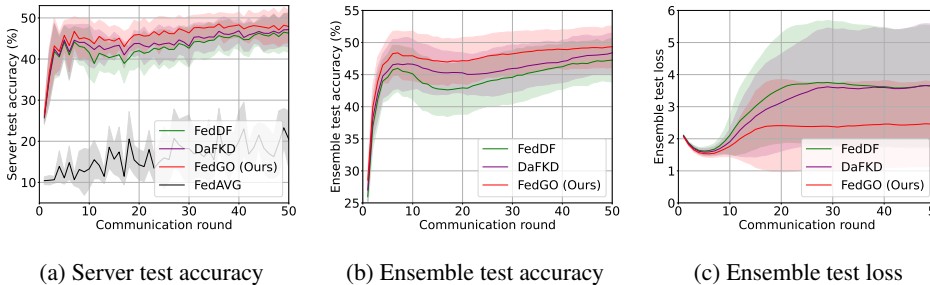

(a) Server test accuracy     (b) Ensemble test accuracy     (c) Ensemble test loss

Figure 10: Test accuracy of server model (%), ensemble test accuracy (%), and test loss of ensemble model of the data-free FedGO with a generator trained from FedGAN (the case (G3)+(D3) of Table 1) and baselines with 100 clients on the CIFAR-10 dataset with $\alpha = 0.05$.

even with pseudo-labels of similar quality, the performance of the server model can decline as the server dataset size decreases. This can be interpreted as the server model becoming more prone to overfitting as the distillation dataset becomes smaller (Hinton, 2015). Note that FedGO has the performance improvement of about 15% over FedAVG even with only 20% of the dataset, which corresponds to 20% of the total client dataset size.

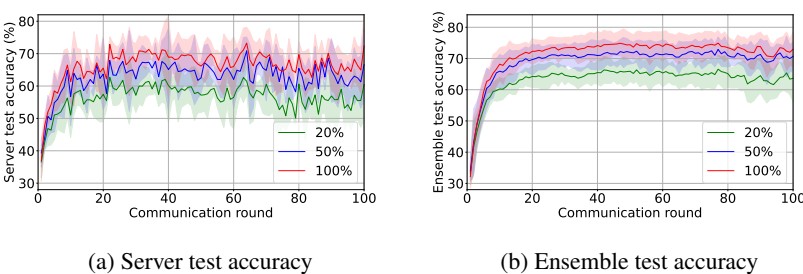

(a) Server test accuracy        (b) Ensemble test accuracy

Figure 11: Server test accuracy (%) and ensemble test accuracy (%) of our FedGO on the CIFAR-10 dataset with $\alpha = 0.05$, according to the size of the unlabeled dataset at the server. In the legend, $X\%$ means that the size of the unlabeled dataset at the server is reduced to $X\%$ of the size assumed in our main CIFAR-10 setting.

### F.5.2 SERVER MODEL TRAINING EPOCHS

Table 8 shows the impact of server model training epochs on FedGO's performance on CIFAR-10 with $\alpha = 0.1$ after 100 communication rounds. Using 5 epochs outperforms 1 epoch, with minimal performance differences beyond 5 epochs. Notably, even with only 1 epoch, FedGO significantly outperforms all the baselines trained with 10 server epochs in Table 2.

Table 8: Server test accuracy (%) and ensemble test accuracy (%) of FedGO on CIFAR-10 with $\alpha = 0.1$ after 100 communication rounds, according to the number of server model training epochs.

| Epoch | 1 | 5 | 10 | 20 |
|---|---|---|---|---|
| Server Test Accuracy | 74.03±6.41 | 79.56±5.30 | 79.62±4.36 | 78.32±5.13 |
| Ensemble Test Accuracy | 77.16±0.88 | 80.97±0.87 | 81.56±0.48 | 81.39±0.75 |

### F.5.3 SERVER MODEL LEARNING RATE DECAY

In the main paper, we used cosine learning rate decay by following the experimental setting of FedDF. As shown in Table 9, the absence of learning rate decay results in further performance improvement. Specifically, an ensemble test accuracy of 85.20% is achieved, which is comparable

to the central training model's accuracy of 85.33%, demonstrating the effectiveness of our provably near-optimal weighting method.

Table 9: Server test accuracy (%) and ensemble test accuracy (%) of FedGO on CIFAR-10 with $\alpha = 0.1$ after 100 communication rounds, with and without learning rate decay during server model training.

|  | FedGO | |
|---|---|---|
|  | with LR decay | without LR decay |
| Server Test Accuracy | 79.62±4.36 | 80.18±2.16 |
| Ensemble Test Accuracy | 81.56±0.48 | 85.20±1.33 |

### F.6 IMPACT OF GENERATOR AND DISCRIMINATOR QUALITY ON PERFORMANCE

#### F.6.1 GENERATOR TRAINING STEPS

Table 10 shows the performance of our FedGO with varying generator training steps (100,000 in the main setup) alongside baseline algorithms after 50 communication rounds, while keeping all other settings unchanged from the main setup. FedGO with the generator trained for 25,000 steps performs better than that with the randomly initialized generator (0 steps), with little performance improvement beyond 25,000 steps. Remarkably, even a randomly initialized generator outperforms FedDF with uniform weighting and achieves performance comparable to DaFKD with a generator trained for 100,000 steps.

Table 10: Server test accuracy (%) and ensemble test accuracy (%) of FedGO on CIFAR-10 with $\alpha = 0.1$ after 50 communication rounds, according to the number of generator training steps.

|  | FedDF | DaFKD | FedGO (ours) | | | | |
|---|---|---|---|---|---|---|---|
| Generator Training Steps | - | 100,000 | 0 | 25,000 | 50,000 | 75,000 | 100,000 |
| Server Test Accuracy | 70.18 ± 2.56 | 71.42 ± 3.11 | 71.12 ± 2.07 | 76.74 ± 3.16 | 78.43 ± 0.99 | 78.89 ± 1.55 | 78.24 ± 1.61 |
| Ensemble Test Accuracy | 73.55 ± 2.41 | 74.54 ± 2.80 | 74.88 ± 1.63 | 79.12 ± 1.97 | 80.72 ± 0.75 | 80.87 ± 0.98 | 80.82 ± 0.82 |

#### F.6.2 DISCRIMINATOR TRAINING EPOCHS

Table 11 shows the final performance of the FedGO algorithm for different numbers of discriminator training epochs on CIFAR-10 with $\alpha = 0.05$. It can be seen that training the discriminator more times results in better final performance. Additionally, we note that among the baselines in Table 2 and Figure 2, except DaFKD which originally trains the generator and discriminators at each round, the highest performance is achieved by the variance weighting method, with the test accuracy of 67.51±10.77%, indicating that there is a performance gain from the FedGO algorithm with just 5 epochs of discriminator training.

Table 11: Server test accuracy (%) of FedGO on CIFAR-10 with $\alpha = 0.05$ at the 100-th communication round, according to the number of discriminator training epochs at the clients.

| Epoch | 1 | 5 | 10 | 30 | 50 |
|---|---|---|---|---|---|
| Accuracy | 63.96±9.03 | 71.38±7.76 | 70.84±8.88 | 72.35±9.01 | **76.92**±5.08 |

#### F.6.3 DISCRIMINATOR ARCHITECTURES

Table 12 presents the number of parameters, the number of FLOPs required for the forward computation, and the performance of FedGO on CIFAR-10 with $\alpha = 0.1$ at the 100-th communication round, when the following three different client discriminator structures are used:

- CNN: The baseline architecture used in the main setting. It consists of four convolutional layers.

- CNN+MLP: A variation of the CNN architecture, where the last two convolutional layers in the CNN are replaced by a single multi-layer perceptron (MLP) layer, resulting in a three-layer shallow network.
- ResNet: A deeper architecture based on ResNet-8, an 8-layer residual network.

Table 12: Server test accuracy (%) of FedGO on CIFAR-10 with $\alpha = 0.1$ at the 100-th communication round along with the number of parameters and the number of FLOPs for the forward computation, according to different client discriminator structures.

| | FedGO | | |
| Discriminator Structure | CNN | CNN+MLP | ResNet |
| --- | --- | --- | --- |
| Number of Parameters | 662,528 | 142,336 | 1,230,528 |
| FLOPs | 17.6 MFLOPs | 9.18 MFLOPs | 51.1 MFLOPs |
| Server Test Accuracy | 79.62±4.36 | 79.71±4.71 | 78.73±5.03 |

Table 12 shows almost identical performances regardless of client discriminator architectures, demonstrating the robustness of FedGO to the discriminator architecture. In particular, the CNN+MLP discriminator, which has less than a quarter of the parameters and around the half of the FLOPs compared to the original CNN structure, achieves similar performance.

## G   COMPREHENSIVE ANALYSIS OF COMMUNICATION, PRIVACY, AND COMPUTATIONAL COMPLEXITY

Let us provide a detailed explanation for Table 1. If the server dataset is available from the outset (first two rows in Table 1), we only need one-shot communication of generator (from the server to the clients) and discriminators (from the clients to the server). Hence, additional communication burden and client-side privacy leakage are negligible. In particular, for our experiments, the parameters of the ResNet-18 classifier are approximately 90MB when stored as a PyTorch state_dict. In comparison, the generator and discriminator models are 4.61MB and 2.53MB, respectively. Over 100 communication rounds, during which ResNet-18 is transmitted repeatedly, the additional communication burden introduced by FedGO is nearly negligible. However, the server dataset is used for distillation for each communication round, incurring non-negligible privacy leakage on the server side. If there is no server dataset (last two rows in Table 1), there is no additional privacy leakage on the server side. To train a generator through FL (last row), multiple rounds of GAN exchanges between the server and clients are required, leading to non-negligible increase in communication burden, client-side privacy leakage and computational burden. If we use a pretrained generator instead (third row), additional communication burden, client-side privacy leakage and computational burden become negligible, but it is challenging in general to secure an off-the-shelf generator which generates data with a distribution similar to the client data distribution.

In the following, we provide a quantitative analysis of additional privacy leakage of FedGO compared to FedAVG, and an explicit comparison of computational cost for FedGO and baselines.

### G.1   PRIVACY ANALYSIS

For privacy measure, we consider local differential privacy (LDP) Dwork et al. (2006) which is widely accepted both in academia and industry. Note that when the data is provided $n$ times by independently applying an LDP mechanism with privacy budget $\epsilon$ for each provision, the total privacy budget becomes $n\epsilon$ from the parallel composition result (Dwork et al., 2014).

Let $T$ denote the total number of communication rounds in the main-FL stage. For the case (G3) in Section 3.2, let $T'$ denote the total number of communication rounds to train a GAN in the pre-FL stage. For simplicity, we assume that every client participates in FL for each communication round. Let $\epsilon_M, \epsilon_D$, and $\epsilon_G$ denote the privacy budgets of LDP mechanisms applied to the classifier, discriminator, generator sent from each client at each communication round, respectively. Let $\hat{\epsilon}_M$ and $\hat{\epsilon}_G$ denote the privacy budgets of LDP mechanisms applied to the classifier and the generator

sent from the server when the server uses its own dataset in case of (S1) for training the generator and for distillation, respectively.

Table 13 shows the client-side and the server-side total privacy leakage of FedAVG and FedGO under various scenarios. For FedAVG, each client provides the classifier $T$ times, and hence the client-side total privacy leakage becomes $T \cdot \epsilon_M$. Let us first analyze the additional client-side privacy leakage of FedGO under various scenarios. For FedGO with the method (G1)+(D1), (G2)+(D1), or (G2)+(D2), the client sends its discriminator only once, incurring extra privacy leakage of $\epsilon_D$, which is negligible with large $T$. For FedGO with (G3)+(D3), the clients need to send the discriminator and the generator for $T'$ times to train a GAN in the pre-FL stage, leading to a non-negligible additional privacy leakage of $T' \cdot (\epsilon_D + \epsilon_G)$ compared to other FedGO scenarios. Next, server-side privacy issues arise only when the server has its own dataset. If the server trains the generator from its dataset and provides it to the clients for the case of (G1), it yields the privacy leakage of $\hat{\epsilon}_G$. In addition, if the server uses its dataset for distillation and applies an LDP mechanism with privacy budget $\hat{\epsilon}_M$ to the classifier for each communication round for the case of (D1), it results in a non-negligible amount of additional privacy leakage $T \cdot \hat{\epsilon}_M$.

Table 13: Quantitative analysis of the client-side and the server-side total privacy leakage of FedAVG and FedGO under various scenarios.

|  |  | Client-side | Server-side |
|---|---|---|---|
| FedAVG |  | $T \cdot \epsilon_M$ | $-$ |
| FedGO | (G1)+(D1) | $T \cdot \epsilon_M + \epsilon_D$ | $\hat{\epsilon}_G + T \cdot \hat{\epsilon}_M$ |
|  | (G2)+(D1) | $T \cdot \epsilon_M + \epsilon_D$ | $T \cdot \hat{\epsilon}_M$ |
|  | (G2)+(D2) | $T \cdot \epsilon_M + \epsilon_D$ | $-$ |
|  | (G3)+(D3) | $T' \cdot (\epsilon_D + \epsilon_G) + T \cdot \epsilon_M + \epsilon_D$ | $-$ |

## G.2 COMPUTATIONAL COST COMPARISON

Table 14 shows the floating point operations (FLOPs) during CIFAR-10 training for the baselines and FedGO with the four scenarios described in Table 1. 1 MFLOP represents $10^6$ FLOPs.

First, on the client side, the computational cost for FedGO with (G1) or (G2) is comparable to that of FedAVG and FedDF, which only optimize the client's vanilla supervised loss. The cost is roughly half of the cost of FedGKD$^+$, which includes a regularization term in the client objective. This reduction is because the client only needs to train the discriminator only once during the pre-FL stage. In each round, FedAVG and FedDF compute 4.17e+7 MFLOPs per client update, whereas the computational cost for training a client's discriminator is 3.29e+7 MFLOPs—less than the cost for one round of classifier training. The additional computation cost for FedGO with (G1) or (G2) is therefore minimal, especially considering its fast convergence speed.[2] Note that the client-side computational cost of FedProx is same as that of FedAVG because the proximal term computation, 1.07e+10, is negligible.

However, in FedGO with (G3), clients train the generator using an FL approach during the pre-FL stage, leading to a significant computational cost on the client side. The same applies to DaFKD, which also trains a generator through an FL approach. The slight difference between DaFKD and FedGO with (G3) is due to one additional step of training client's discriminator in the pre-FL stage of FedGO. However, as the computational and communication capabilities of devices continue to improve, many recent studies like those referenced in the main paper are actively exploring data-free FL approaches.

Next, on the server side, in FedGO with (G1)+(D1), training a generator using server dataset involves significant additional computation compared to FedDF due to the 100,000 steps required for training a ResNet-based generator and discriminator. However, given that federated learning typically involves a server with ample resources and clients with limited computational resources, this increase in server-side computation is more affordable in practice, compared to increasing the

---

[2]We note that the computational cost exceeds 2%, rather than being below 1%, because in each round, only $C = 0.4$ proportion of clients are sampled to participate in federated learning, rather than full client participation.

computational burden on clients. Furthermore, while the computation for training the generator is irrelevant to the number of clients, the computation required for pseudo-labeling scales linearly with the number of clients. Note that the total computational cost in Table 14 assumes 20 clients. In real-world scenarios, where 100+ clients may participate in FL, the relative proportion of the computational cost for training the generator will decrease.

Note that using an off-the-shelf generator reduces the additional server-side computational cost of FedGO. FedGO with (G2)+(D1) requires approximately 2% more computation than FedDF, but achieves a significant performance gain of about 13%p on CIFAR-10 with $\alpha = 0.05$ in Table 2. The reason why FedGO with (G2)+(D2) has a higher server-side computational cost compared to FedGO with (G2)+(D1) is that FedGO with (G2)+(D2) generates distillation dataset using a heavy generator, StyleGAN.

FedGO with (G3)+(D3) also generates distillation dataset using a generator but the generator used here is lighter than StyleGAN generator used in (G2)+(D2). The computational cost for the generation of distillation dataset in FedGO with (G3)+(D3) is 2.11e+7 MFLOPs which is negligible compared to the computational cost for pseudo-labelling and ensemble distillation which is 5.07e+10 MFLOPs. On the other hand, note that the computational cost of FedGO with (G3)+(D3) is slightly lower than DaFKD while both train a generator using an FL approach. The reduction mainly comes from the difference that FedGO with (G3)+(D3) generates a distillation dataset only once after the training of the generator, while DaFKD updates the distillation dataset in every communication round. Finally, note that the server-side computational cost of FedGO with (G3)+(D3) is comparable to the case (G2)+(D1), even though (G3) trains a generator through an FL approach. This is because the server's role is limited to averaging the clients' generator and discriminator, incurring negligible additional computational cost on the server side.

Table 14: The number of MFLOPs for training our FedGO and baselines on CIFAR-10 for 100 communication rounds.

| | FedAVG | FedProx | FedDF | FedGKD$^+$ | DaFKD | FedGO (ours) | | | |
|---|---|---|---|---|---|---|---|---|---|
| | | | | | | (G1)+(D1) | (G2)+(D1) | (G2)+(D2) | (G3)+(D3) |
| Client-side | 3.33e+10 | 3.33e+10 | 3.33e+10 | 6.67e+10 | 8.81e+11 | 3.40e+10 | 3.40e+10 | 3.40e+10 | 8.82e+11 |
| Server-side | 7.82e+3 | 7.82e+3 | 5.00e+10 | 5.00e+10 | 5.28e+10 | 1.39e+11 | 5.07e+10 | 6.01e+10 | 5.07e+10 |
| Total | 3.33e+10 | 3.33e+10 | 8.33e+10 | 1.17e+11 | 9.34e+11 | 1.73e+11 | 8.47e+10 | 9.41e+10 | 9.32e+11 |

## H    LIMITATION

Our study does not provide specific guidance on the selection of discriminator architectures, which may affect the overall performance of the federated learning system. Additionally, although our FedGO algorithm can be extended to model heterogeneous scenarios as in FedDF, we found it challenging to define an optimal model ensemble for multiple hypothesis classes. Consequently, it appears difficult to apply the results of Theorem 2 and Corollary 1 in such cases.

## I    BROADER IMPACTS

In this work, we proposed a federated learning algorithm that demonstrates strong performance in scenarios where client data is heterogeneous. This capability makes our approach highly effective for distributed learning in many practical situations, where data across different clients can vary significantly. By efficiently handling such data diversity, our algorithm holds the potential to enhance the applicability and robustness of federated learning systems in real-world applications.

