# OpenReview forum: "FedGO : Federated Ensemble Distillation with GAN-based Optimality"
_ICLR.cc/2025/Conference — Submitted to ICLR 2025_

### Official Review · Reviewer_KVTQ · 2024-11-03

**Soundness:** 2
**Presentation:** 2
**Contribution:** 2
**Rating:** 5
**Confidence:** 4

**Summary:**

This paper proposes FedGO: Ferated Ensemble Distillation with GAN-based Optimality, for federated ensemble distillation. This algorithm incorporates a novel weighting method using the client discriminators that are trained at the clients based on the generator distributed from the server and their own datasets. The generator distributed from the server can be either off-the-shelf or trained with the unlabeled dataset on the server. The exchange of the generator and the client discriminators between the server and the clients occurs only once before the main FL algorithm starts, resulting in minimal additional overhead.
Extensive experiments demonstrate significant improvements of FedEDG over existing research both in final performance and convergence speed on multiple image datasets.

**Strengths:**

1. The paper is well-written and easy to follow.
2. The authors conducted extensive experiments to verify the effectiveness of the proposed method.

**Weaknesses:**

As far as I am concerned, distillation-based FL is data dependent and requires access to an auxiliary dataset derived from publicly available proxy data sources for knowledge transfer, whereas a desirable auxiliary dataset is not always available since its construction requires careful deliberation and even prior knowledge about clients’ private data to achieve satisfactory performance, which is inconsistent with the privacy-preserving nature of FL. In addition, I argue that FedEDG with Pretrained Generator proposed in this paper also has the above-mentioned issues. This is because pre-trained generator needs to be trained on public datasets. Therefore, I remain skeptical of this research direction, even if the paper contains theoretical evidence. Furthermore, if the author wants to convince me, please provide some feasible solutions to address the aforementioned issues.
I'll raise my score if author can address the above problems.

**Questions:**

Please see Weaknesses.

---

> ### Author Response · Authors · 2024-11-15
>
> First, we would like to express our gratitude for the time and effort you put into reviewing our paper. We agree with the feedback that the use of an unlabeled dataset on the server (no need to share with clients for FedGO) or pretrained generator may not always be feasible. In fact, our paper presents a solution for such situations: we developed a method for our FedGO algorithm to operate in a data-free setting without the need for an additional dataset or pretrained generator, and we included both the method and experimental results in the paper.
>
>
> As outlined in Section 3.2, we propose a data-free approach: when the server does not have a dataset (S2), the generator is trained using FL techniques like FedGAN (G3), and a distillation dataset is created with that generator (D3). The experimental results for this approach are provided in Appendix F.3. Furthermore, we offer a comprehensive comparison and analysis in Appendix G, addressing communication, privacy, and computation aspects for scenarios involving an additional dataset, pretrained generator, or a data-free approach.
>
>
> To summarize, **our paper already proposes a data-free approach that does not require an external dataset or pretrained generator, and it includes experimental results as well as a multi-faceted analysis covering privacy and computation aspects**. We hope that our comprehensive analysis of various scenarios will have a positive impact on your evaluation. Thank you.

---

> > ### Comment · Reviewer_KVTQ · 2024-11-23
> >
> > Thank you for your response. However, your assurance does not align with the statements in your paper. For instance, in Figure 4, I can clearly see that FedGO requires a pre-trained generator (i.e., Generator preparation) and the server dataset. Moreover, FedGO necessitates the pre-training of additional client discriminators using local private data on the client side, which introduces additional computational and memory costs for the deployment of FedGO. This is because, in actual FL scenarios, the computational and memory resources on the client side are scarce. Therefore, I maintain a skeptical attitude towards this work and keep my current score unchanged.

---

> ### Author Response · Authors · 2024-11-25
>
> We appreciate your detailed feedback and would like to clarify a potential misunderstanding regarding the data-free approach proposed in our work. In the data-free scenario (G3)+(D3), **FedGO does not require a pre-existing server dataset or pretrained generator before the pre-FL stage**. Instead, both **the generator and the distillation dataset are constructed during the pre-FL stage using a fully data-free methodology**.  Additionally, while client-side resources are used to train discriminators, **the computational and memory overhead has been carefully evaluated to be minimal,** as detailed below.
>
> 1. **Server Dataset and/or Pretrained Generator**
>
>     As outlined in Section 3.2 of our paper, under the data-free scenario (G3) + (D3), the generator is trained using FL techniques, such as FedGAN. This approach **does not require any public dataset, unlabeled data available only on the server, or prior knowledge of client data**. The generator then produces synthetic data, which is used for ensemble distillation.
>
>     To further clarify, when we state that a pretrained generator is not required in the data-free scenario, we specifically contrast this with scenarios like (G2) in Table 1 of Section 3.2. In (G2), a pretrained generator (e.g., StyleGAN trained on large, public datasets) is necessary, and this generator must be available prior to the pre-FL stage. In contrast, **our data-free approach (G3) avoids this requirement entirely by training the generator dynamically within the FL framework**, thereby eliminating the dependency on large external datasets or pretrained models.
>
> 2. **Clarification of Figure 4**
>
>     Fig. 4 illustrates that the generator is prepared according to one of the three methods, (G1, G2, or G3). Here, **(G3) represents the previously mentioned data-free approach**, which does not require any public dataset, unlabeled data available only on the server, or prior knowledge of client data.
>
> 3. **Client-Side Resources**
>
>     We acknowledge your concern regarding the additional computational overhead introduced by the pre-training of client discriminators, especially in resource-constrained FL scenarios.
>
>     Our FedGO algorithm incorporates a provably near-optimal weighting method, which minimizes additional client-side computational overhead. To support this, we provided extensive analyses of the communication, privacy, and computational costs in Section 3.2, Table 1, Appendix G, and Table 9 of the submitted paper.
>
>     Specifically, for scenarios other than the data-free case (G3) + (D3), Table 9 shows that FedGO incurs only approximately 2% additional client-side computational cost compared to FedAvg/FedDF. In the (G2) + (D1) scenario, it imposes less than 1.5% additional server-side computational cost compared to FedDF. These results demonstrate that FedGO operates efficiently while maintaining strong performance.
>
>     Additionally, during this rebuttal phase, we conducted further experiments on the performance of the FedGO algorithm using different discriminator architectures. We found that even with a client discriminator structure that has less than 1/4 the number of parameters and forward FLOPs of the discriminator used in the initially submitted paper, FedGO still achieves nearly identical performance. This result will be included in the revised paper. These results indicate that FedGO may require even less additional client computation and memory overhead than the results reported in Table 9 (which were already very small), while maintaining the same server model performance.
>
>
> We hope this explanation resolves your concerns about the necessity of server datasets or pretrained generators in the data-free scenario. The flexibility of FedGO to operate effectively in both data-free and auxiliary-data settings ensures its adaptability to various FL environments, including those with limited resources or stringent privacy requirements.
>
> Thank you for your continued engagement. We are happy to address any further questions or concerns you may have.

---

> ### Author Response · Authors · 2024-11-29
>
> We are glad that some of your concerns were resolved through our additional experiments and explanations, and we appreciate the opportunity to address your remaining questions about computational costs and privacy risks.
>
> Federated ensemble distillation has emerged as **a powerful approach for addressing challenges like data heterogeneity, with numerous studies presented at major conferences reflecting its value and relevance in the research community**. For example, FedDF is an early study on federated ensemble distillation, presented at NeurIPS 2020. It has been **cited over 1,000 times to date, reflecting the academic community's significant interest in and recognition of this methodology**. One central aspect of this framework is the requirement for a distillation dataset, which can either pre-exist on the server or be constructed dynamically using data-free methods. Both directions have been actively explored.
>
> In our work, we propose a method that operates within the same federated ensemble distillation scenario as prior approaches. As demonstrated in Appendix G, FedGO introduces **only minimal computational overhead and privacy leakage on the client side,** compared to other federated ensemble distillation methods such as FedDF. With this minimal additional overhead, **our method benefits from a theoretically guaranteed, provably near-optimal weighting scheme, thereby achieving state-of-the-art performance in federated ensemble distillation**. Furthermore, this approach enables us to reach the same level of performance with fewer communication rounds, making our algorithm **more efficient in terms of client-side computational, communication, and privacy costs**, thanks to faster convergence.
>
> Therefore, we believe that **concerns regarding computational cost or privacy risks should not lead to an undervaluation of our work.** To do so would risk dismissing the broader progress made in federated ensemble distillation, which has been actively and extensively studied in the research community. We hope this context provides clarity and addresses any remaining concerns.

---

### Official Review · Reviewer_5yYt · 2024-11-03

**Soundness:** 3
**Presentation:** 3
**Contribution:** 3
**Rating:** 8
**Confidence:** 4

**Summary:**

The paper introduces a new approach to address the issue of data heterogeneity in federated learning. By applying Generative Adversarial Network (GAN) techniques to federated ensemble distillation, the paper proposes a near-optimal weighting method that enhances the training process of the server model. Extensive experimental validation demonstrates significant improvements in model performance and convergence speed across various image classification tasks. Moreover, the study provides an in-depth analysis of the potential additional communication costs, privacy leaks, and computational burdens introduced by this method, showcasing its practicality and flexibility in protecting data privacy and enhancing system efficiency.

**Strengths:**

This paper demonstrates originality through its innovative integration of GAN-based techniques with federated ensemble distillation. The use of discriminators trained at the client side to optimize the weighting of client contributions during the distillation process is a novel approach that has not been extensively explored in previous federated learning research.

The method's originality is further enhanced by its theoretical grounding, which employs results from GAN literature to develop a provably near-optimal weighting method.

The experimental setup is well thought out

**Weaknesses:**

The paper claims near-optimal performance based on theoretical justifications rooted in GAN literature. However, these claims might depend heavily on certain idealized assumptions about data distributions and discriminator performance. Real-world deviations from these assumptions could lead to suboptimal performance. The paper does not explain how to select discriminator architectures.

**Questions:**

1. In introduction on page 2, Our main contributions are summarized in the following: "Federated Ensemble Distillation" instead of "Ferated Ensemble Distillation".

2. In theoretical analysis, near-optimal performance is heavily affected on discriminator performance. I do not understand how to select the discriminator architectures? Can you give me some detailed description?

---

> ### Author Response · Authors · 2024-11-27
>
> Thank you for reviewing our paper and recognizing the value of our results. We also appreciate you pointing out the typo in the introduction; we have corrected it in the revised paper.
>
> We greatly appreciate your question about the selection of discriminator architectures. To address it,  we have conducted experiments with the following three different client discriminator architectures:
>
> - **CNN**: The baseline architecture used in the submitted paper. It consists of four convolutional layers.
> - **CNN+MLP**: A variation of the CNN architecture, where the last two convolutional layers in the CNN are replaced by a single multi-layer perceptron (MLP) layer, resulting in a three-layer shallow network.
> - **ResNet**: A deeper architecture based on ResNet-8, an 8-layer residual network.
>
> The table below summarizes the number of parameters, the number of forward computation FLOPs, and the server model's test accuracy  on CIFAR-10 with $\alpha = 0.1$  at the 100-th communication round when using these three different discriminator architectures.
>
> \\begin{array}{|l|ccc|}
> \\hline
> &&\\text{FedGO}&\\\\
> \\text{Discriminator Sturcture} & \\text{CNN} & \\text{CNN+MLP} & \\text{ResNet} \\\\
> \\hline
> \\text{Number of Parameters} & 662,528 & 142,336 & 1,230,528\\\\
> \\text{MFLOPs} & 17.6 & 9.18  & 51.1 \\\\
> \\text{Server Test Accuracy} & 79.62 \pm 4.36 & 79.71 \pm4.71 & 78.73 \\pm 5.03 \\\\
> \\hline
> \\end{array}
>
> As seen in the table above, all discriminator architectures achieve nearly identical server model performance. These results demonstrate that the performance of FedGO algorithm is robust to different discriminator architectures and maintains strong performance regardless of the chosen structure.
>
> Therefore, we recommend the CNN+MLP discriminator, as it significantly reduces client-side computation and memory overhead while delivering competitive results. This flexibility enables FedGO to effectively adapt to diverse FL scenarios with varying resource constraints.
>
> We hope this response clarifies your concerns and highlights the adaptability of our approach. Thank you again for your constructive feedback. We look forward to addressing any additional questions you might have.

---

### Official Review · Reviewer_Gtch · 2024-11-04

**Soundness:** 2
**Presentation:** 2
**Contribution:** 3
**Rating:** 5
**Confidence:** 4

**Summary:**

This paper proposed a novel federated ensemble distillation approach that utilizes generative adversarial networks (GANs) to address the challenges posed by data diversity across clients. Specifically, the proposed approach employs GANs to optimize the weighting of client predictions, thereby improving the quality of pseudo-labels generated during the ensemble distillation process. The paper provides theoretical insights that establish the effectiveness of the proposed method. Comprehensive experiments demonstrate that the proposed approach outperforms existing methods in robustness against data heterogeneity.

**Strengths:**

1. The paper provides a theoretical foundation for the proposed approach, which validates its effectiveness and enhances its credibility.
2. The paper analyzes communication, privacy, and computational complexity within different scenarios, providing valuable insights for implementing the proposed approach.

**Weaknesses:**

1. This paper needs to demonstrate the effectiveness of the proposed approach on different model structures, such as VGG and MobileNet.
2. The effectiveness of the proposed method relies on the quality of the discriminator and generator. The paper needs to conduct related ablation studies.
3. This paper should conduct ablation studies to analyze the impact of hyperparameters (e.g. $E_s$ and $E_d$) on the effectiveness of the approach.
4. The experimental settings of the baselines are not clearly stated, and it is important to clarify the fairness of the experimental comparison.
5. The additional computational and communication overhead introduced by the GAN-based approach may not be suitable for FL scenarios, particularly those with strict resource constraints.

**Questions:**

Please refer to the weaknesses.

---

> ### Author Response · Authors · 2024-11-21
>
> Thank you very much for taking the time to read and review our paper. In accordance with the reviewer’s comments, we conducted several additional experiments on CIFAR-10 with 𝛼=0.1. The experimental results were obtained using five different random seeds, and the reported results are presented as the mean ± standard deviation.
>
> 1. **Different Model Structures**
>
> In accordance with the reviewer’s suggestion, we conducted additional experiments using different model structures, which are VGG11 (with BatchNorm Layer) and ResNet-50. For VGG11, both the client and server models were trained using SGD with a learning rate of 0.01 and momentum of 0.9, and all the other settings including hyperparameters were kept identical to those in the initially submitted paper. We implemented VGG11 based on https://github.com/chengyangfu/pytorch-vgg-cifar10. For ResNet-50, all the settings including optimizer and hyperparameters were the same as the initially submitted paper. The table below presents the server test accuracy of central training, FedDF, and FedGO with the aforementioned model structures after 100 communication rounds.
> \\begin{array}{|l|c|c|}
> \\hline
> & \\text{VGG11} & \\text{ResNet-50} \\\\
> \\hline
> \\text{Central training} & 83.27 \\pm 0.60 & 85.12 \\pm 0.44 \\\\
> \\text{FedDF} & 68.59 \\pm 4.65 & 65.21 \\pm 4.62 \\\\
> \\textbf{FedGO (ours)} & 72.53 \\pm 4.10 & 75.52 \\pm 4.30 \\\\
> \\hline
> \\end{array}
> We can see that our FedGO algorithm consistently achieves performance gains over FedDF across different model structures. We will update experimental results for an additional baseline algorithm, FedGKD$^+$, within this rebuttal period, and for all the other baseline algorithms in the final camera-ready version of the paper.

---

> ### Author Response · Authors · 2024-11-21
>
> 2. **Quality of Discriminator and Generator**
>
> We agree with the reviewer on the importance of analyzing the impact of the quality of the discriminator and generator. Note that we already reported the experimental results according to the discriminator quality in Appendix F.5 and Table 7 of the initially submitted paper. Thus, we focused on additional experiments to evaluate the impact of the generator quality.
>
> Keeping all other settings unchanged from our main setup, we measured the performance of our FedGO with varying generator training steps (originally 100,000) alongside baseline algorithms after 50 communication rounds. The results are summarized in the table below:
> \\begin{array}{|l|c|c|ccccc|} \\hline & \\text{FedDF} & \\text{DaFKD} &&& \\textbf{FedGO (ours)}& \\\\
> \\text{Generator Training Steps} & - & \\text{100,000 Steps} & \\text{0 Steps} & \\text{25,000 Steps} & \\text{50,000 Steps} & \\text{75,000 Steps} & \\text{100,000 Steps} \\\\ \\hline \\text{Server Test Accuracy} & 70.18 \\pm 2.56 & 71.42 \\pm 3.11 & 71.12 \\pm 2.07 & 76.74 \\pm 3.16 & 78.43 \\pm 0.99 & 78.89 \\pm 1.55 & 78.24 \\pm 1.61 \\\\
> \\text{Ensemble Test Accuracy} & 73.55 \\pm 2.41 & 74.54 \\pm 2.80 & 74.88 \\pm 1.63 & 79.12 \\pm 1.97 & 80.72 \\pm 0.75 & 80.87 \\pm 0.98 & 80.82 \\pm 0.82 \\\\ \\hline \\end{array}
> As shown in the above table, FedGO with the generator trained for 25,000 steps performs better than that with the randomly initialized generator (0 steps), with little performance improvement beyond 25,000 steps. Remarkably, even a randomly initialized generator outperforms FedDF with uniform weighting and achieves performance comparable to DaFKD with a generator trained for 100,000 steps.
>
> 3. **Impact of Hyperparameters**
>
> Thanks for the comment.  For the discriminator training epochs $E_d$, we already reported relevant experimental results in Appendix F.5 and Table 7 of the initially submitted paper. To address the reviewer’s suggestion, we additionally evaluated the impact of server epochs ($E_s$, set to 10 in the initially submitted paper) on FedGO’s performance after 100 communication rounds.
>
> As shown in the above table, using 5 epochs outperforms 1 epoch, with minimal performance differences beyond 5 epochs. Notably, even with only 1 epoch, FedGO significantly outperforms all the baselines trained with 10 server epochs in the initially submitted paper (Table 2 of the paper).
> \\begin{array}{|l|c|c|c|c|}
> \\hline
>  & \\text{1 Epoch} & \\text{5 Epochs} & \\text{10 Epochs} & \\text{20 Epochs} \\\\
> \\hline
> \\text{Server Test Accuracy (\\%)} & 74.03 \\pm 6.41 & 79.06 \\pm 5.30 & 79.62 \\pm 4.36 & 78.32 \\pm 5.13 \\\\
> \\text{Ensemble Test Accuracy (\\%)} & 77.16 \\pm 0.88 & 80.97 \\pm 0.87 & 81.56 \\pm 0.48 & 81.39 \\pm 0.75 \\\\
> \\hline
> \\end{array}
> Moreover, we conducted an additional experiment to evaluate the impact of the server model’s learning rate decay on FedGO’s performance after 100 communication rounds. In the initially submitted paper, we used  cosine learning rate decay by following the experimental setting of FedDF.
> \\begin{array}{|l|ccc|}
> \\hline
> && \\text{FedGO}&\\\\
> & \\text{with LR decay}&&\\text{without LR decay}\\\\
> \\hline
> \\text{Server Test Accuracy} & 79.62 \\pm 4.36 && 80.18 \\pm 2.16 \\\\
> \\text{Ensemble Test Accuracy} & 81.56 \\pm 0.48 && 85.20 \\pm 1.33 \\\\
> \\hline
> \\end{array}
> As shown in the above table, the absence of learning rate decay resulted in further performance improvement. Specifically, an ensemble test accuracy of 85.20% was achieved, which is comparable to the central training model’s accuracy of 85.33%, demonstrating the effectiveness of our provably near-optimal weighting method.

---

> ### Author Response · Authors · 2024-11-21
>
> 4. **Experimental settings**
>
> We agree that it is important to clearly state the experimental settings of the baselines. As detailed in Appendix E.2 of the initially submitted paper, we have thoroughly documented the experimental settings of our FedGO and baseline methods. For all the baseline experiments, the random seeds, data splits, and model structures were the same as FedGO, and some hyperparameters specific to each baseline algorithm were optimized using grid search to select the best-performing values. If there are additional details that you believe should be reported, please let us know, and we will be happy to include them.
>
> 5. **Additional Overhead due to Utilizing GAN**
>
> We acknowledge the reviewer’s concern that the additional computational and communication overhead introduced by the GAN-based approach could present challenges in FL scenarios, particularly under strict resource constraints. However, given the nature of FL, the server often communicates with numerous clients, aggregates client models, and trains the server model. As such, many studies assume—and often find in practice—that the server typically has significantly more computational and communication resources than clients. Building on this assumption, several federated ensemble distillation studies focus on imposing additional computation on the server, rather than on the clients, to achieve faster convergence for the same communication budget.
>
> Our FedGO algorithm, which is GAN-based, implements a **provably near-optimal weighting method** with **minimal additional client-side computation and communication overhead**. To substantiate this, we provided extensive analyses of the communication, privacy, and computational costs in Section 3.2, Table 1, Appendix G, and Table 9 in the initially submitted paper.
>
> Let us first focus on the scenarios other than the data-free case (G3) + (D3). As shown in Table 9, FedGO imposes only around 2% additional computational cost on the client side compared to FedAVG/FedDF. In particular, in the (G2) + (D1) scenario, it incurs less than 1.5% additional server-side computational cost compared to FedDF. Regarding communication cost, the additional overhead introduced by FedGO is also negligible, compared to FedAVG/FedDF. The only additional communication required is a one-shot exchange of the generator and discriminator between the server and clients. In our experiments, the parameters of the ResNet-18 classifier were approximately 90MB when stored as a PyTorch `state_dict`. In comparison, the generator and discriminator models are 4.61MB and 2.53MB, respectively. Over 100 communication rounds, during which ResNet-18 is transmitted repeatedly, the additional communication introduced by FedGO is nearly negligible.
>
> Furthermore, the total computational cost in Table 9 assumes 20 clients. On the server side, the most computationally demanding case for utilizing FedGO on the server side occurs when training the generator in the (G1)+(D1) scenario, which accounts for approximately 63.5% of the total computational cost.  However, once the generator is trained in the pre-FL stage, no further computation is needed. In real-world scenarios, where 100+ clients may participate in FL, the computation required for pseudo-labeling scales linearly with the number of clients. Consequently, as the number of clients increases, the relative proportion of the computational cost for training the generator decreases. Additionally, as shown in Figure 3, FedGO demonstrates significant performance advantages and faster convergence rates compared to baseline algorithms as the number of clients increases. This suggests that, in terms of computational and communication cost efficiency, **FedGO may be a more effective algorithm for achieving the same performance**.
>
> For the data-free FedGO with (G3)+(D3), we recognize that such data-free approaches impose non-negligible communication and computational costs on both the client and server sides, potentially limiting their applicability in resource-constrained environments. However, as the computational and communication capabilities of devices continue to improve, many recent studies—like those referenced in our initially submitted paper—are actively exploring data-free FL approaches. In this context, our study aligns with the growing body of research pushing the boundaries of FL capabilities while addressing modern hardware advancements. We believe our work contributes meaningfully to this evolving field and offers a promising avenue for further exploration.

---

> > ### Comment · Reviewer_Gtch · 2024-11-25
> >
> > Thanks for the reply!
> >
> > The additional experiments addressed some of my concerns. However, I'm still concerned about the extra overhead of FedGO.
> > We acknowledge that FedGO has excellent performance and satisfactory additional overhead in scenarios other than the data-free case (G3) + (D3). However, the extra overhead of FedGO in data-free scenarios is impossible to ignore, which greatly reduces the usability of FedGO because we mainly focus on data-free scenarios when discussing federated learning.
> >
> > Therefore, I finally decided to raise my score to 5.

---

> ### Author Response · Authors · 2024-12-04
>
> Thank you for your detailed review and for highlighting concerns about the additional computation overhead in the data-free (G3)+(D3) scenario. We appreciate the opportunity to address this important point.
>
> We conducted additional experiments to validate the effectiveness of data-free FedGO when significantly reducing computation overhead. To reduce computation overhead, we used smaller structures for the GAN (to train the generator via FL) and the client-side discriminator (for weighting after generator training), and reduced the number of local epochs and the number of communication rounds for training the generator in a data-free manner. Specifically, for the GAN, we utilized a simplified DCGAN structure based on [this implementation](https://github.com/Ksuryateja/DCGAN-MNIST-pytorch/blob/master/gan_mnist.py), modifying the number of channels to 3. For the client-side discriminator, we adopted the CNN+MLP architecture described in our earlier response. Furthermore, unlike the submitted paper where the GAN was trained in the pre-FL stage with 30 local epochs and 100 communication rounds, we prepared the GAN in this experiment using only 5 local epochs and 5 communication rounds. Then, we compared the performance of FedGO after 50 communication rounds on CIFAR-10 with $\alpha = 0.1$ for 100 clients, to other data-free FL algorithms FedAVG, FedProx, FedGKD, SCAFFOLD [1], and FedDisco [2]. The last two baselines have been newly added to ensure a comprehensive comparison.
>
> \\begin{array}{|l|c|c|c|c|c|c|}
> \\hline
> & & & & && \\text{FedGO} \\\\
>  & \\text{FedAVG} & \\text{FedProx} & \\text{Scaffold} & \\text{FedGKD} & \\text{FedDisco} & \\text{(G3)+(D3)}\\\\
> \\hline
> \\text{Server Test Accuracy} & 33.96 \\pm 4.20 & 36.80 \\pm 3.96 & 37.94 \\pm 2.73 & 37.2 \\pm 3.21 & 36.53 \\pm 2.96 & \\textbf{40.45} \\pm 4.77 \\\\
> \\text{Client-side MFLOPs} & 1.667 \\text{e+10} & 1.667 \\text{e+10} & 1.667 \\text{e+10} & 3.336 \\text{e+10} & 1.667 \\text{e+10} & 1.671 \\text{e+10} \\\\
> \\hline
> \\end{array}
>
> As shown in the table above, **FedGO achieves superior performance compared to baseline algorithms while maintaining minimal additional client-side computation.** Specifically, the MFLOPs required to prepare the generator and discriminator are only **4.682e+7**, which is less than **3%** of the MFLOPs required for classifier training over 100 communication rounds (**1.666e+10**).
>
> This efficiency is achieved through the use of a significantly smaller GAN structure, demonstrating that **even with minimal computational and communication requirements, our approach delivers notable performance improvements**. We hope this addresses your concerns and highlights the practical viability of FedGO in the data-free (G3)+(D3) scenario.
>
> [1] Karimireddy, Sai Praneeth, et al. "Scaffold: Stochastic controlled averaging for federated learning." *International conference on machine learning*. PMLR, 2020.
>
> [2] Ye, Rui, et al. "Feddisco: Federated learning with discrepancy-aware collaboration." *International Conference on Machine Learning*. PMLR, 2023.

---

### Author Response · Authors · 2024-11-29

We have uploaded a revised version of the paper, incorporating additional experiments and analyses conducted during the rebuttal period. All changes from the initially submitted paper are highlighted in blue. A summary of major changes is provided below:

1. **Additional Experiments with Alternative Architectures**

    **Classifier Architectures:** We conducted the main experiments using two different classifier architectures. The results demonstrate that FedGO consistently outperforms other baseline algorithms regardless of the model architecture. Details can be found in Appendix F.3.

    **Discriminator Architecture:** We assessed the impact of discriminator architectures on FedGO's performance. The results indicate that FedGO achieves similar final performance regardless of the discriminator architecture. These findings are summarized in Appendix F.6.

2. **Analysis of Impact of Hyperparameters**

    **Server Model Training Epochs:** We evaluated the performance of FedGO with varying numbers of training epochs for the server model. The experiments show that FedGO achieves higher performance than baseline algorithms even with fewer server model training epochs. This analysis is detailed in Appendix F.5.

    **Learning Rate Decay:** We investigated the effect of learning rate decay during server model training. The results reveal that FedGO performs better without learning rate decay. This is also discussed in Appendix F.5.

    **Generator Training Epochs:** We examined the effect of varying the training epochs of the generator on FedGO's performance. Interestingly, even when using an untrained, randomly initialized generator, FedGO outperforms or matches the performance of baseline algorithms. Further details are available in Appendix F.6.

3. **Additional Explanation on the Overhead of FedGO**

    We have provided a more detailed explanation emphasizing that the additional communication and computation overhead of our FedGO is minimal compared to previous federated ensemble distillation methods.


These updates aim to further clarify and strengthen the findings of our work.

---

### Author Response · Authors · 2024-12-04

Thank you for acknowledging that our weighting method for federated ensemble distillation is a novel approach with theoretically guaranteed optimality. However, concerns regarding additional costs due to the use of GAN have been raised. As we have explained in detail to the first and third reviewers, we would like to reaffirm that these additional costs are not a significant concern.

As detailed in Appendix G and our most recent response to the first reviewer (**Gtch)**, **FedGO incurs only negligible additional computational and communication costs on the client side, even in a fully data-free setup**. Despite this minor overhead, FedGO leverages a theoretically guaranteed, provably near-optimal weighting approach, enabling it to achieve state-of-the-art performance in federated ensemble distillation. Furthermore, it achieves comparable performance with fewer communication rounds, reducing client-side computation, communication, and privacy overheads through faster convergence.

We respectfully assert that concerns about additional costs should not detract from the substantial contributions our work makes to the field of federated ensemble distillation.

---

### Meta-Review · Area_Chair_QbY1 · 2024-12-22

**Metareview:**

Summary: The paper introduces FedGO, a federated learning approach that uses Generative Adversarial Networks (GANs) to optimize client prediction weighting in the ensemble distillation process, aiming to improve robustness against data heterogeneity. The approach provides theoretical insights and shows improved performance and convergence speed in image classification tasks. However, concerns remain about the additional computational overhead and privacy implications of FedGO.

Strengths:

FedGO offers a novel integration of GAN techniques with federated ensemble distillation, potentially enhancing the training process of server models.

The paper provides a theoretical foundation for the method and demonstrates its effectiveness through extensive experiments.
Drawbacks:

The approach may introduce significant additional computational and communication overhead due to the need for discriminator training and uploading, which could be prohibitive in resource-constrained federated learning scenarios.

There are concerns about the privacy implications of FedGO, as it requires the upload of locally trained discriminators, increasing the risk of privacy leakage.

The effectiveness of FedGO relies heavily on the quality of the discriminator and generator, and the paper lacks ablation studies to analyze the impact of hyperparameters and the robustness of the method under different conditions.

Given the above points, I must reject this work due to concerns about its practical applicability, the potential increase in privacy risks, and the need for more comprehensive analysis to address the drawbacks identified.

**Additional Comments On Reviewer Discussion:**

Concerns are not well-addressed.

---

### Decision · Program_Chairs · 2025-01-22

Reject